# TOWARDS SEQUENCE MODELING ALIGNMENT BETWEEN TOKENIZER AND AUTOREGRESSIVE MODEL

**Pingyu Wu**[1,2,*], **Kai Zhu**[1,2,*], **Yu Liu**[2], **Longxiang Tang**[2]
**Jian Yang**[1], **Yansong Peng**[1,2], **Wei Zhai**[1], **Yang Cao**[1], **Zheng-Jun Zha**[1,†]

[1] University of Science and Technology of China    [2] Tongyi Lab
wpy364755620@mail.ustc.edu.cn

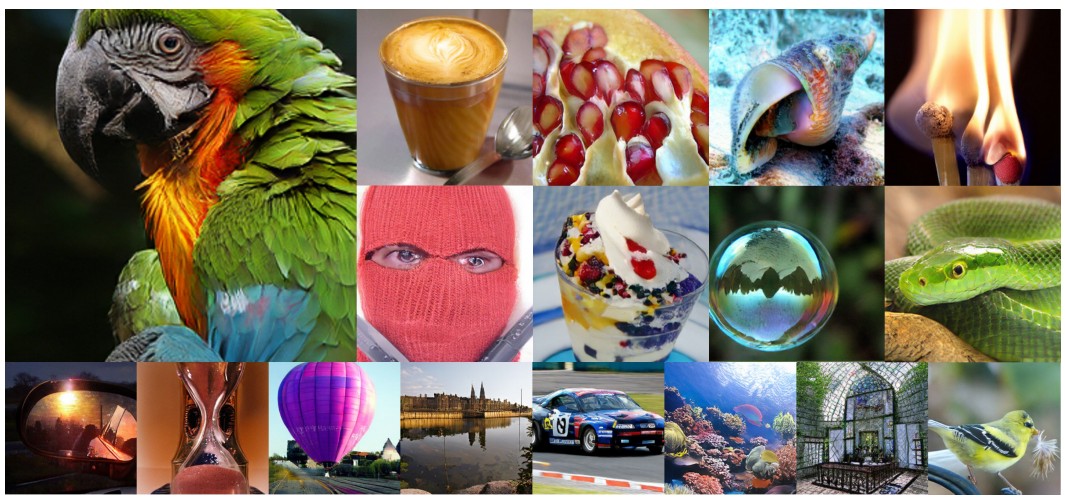

Figure 1: **256×256 samples** of class-conditional generation on ImageNet using our AliTok-XL model (662M).

## ABSTRACT

Autoregressive image generation aims to predict the next token based on previous ones. However, this process is challenged by the bidirectional dependencies inherent in conventional image tokenizations, which creates a fundamental misalignment with the unidirectional nature of autoregressive models. To resolve this, we introduce AliTok, a novel **Ali**gned **Tok**enizer that alters the dependency structure of the token sequence. AliTok employs a bidirectional encoder constrained by a causal decoder, a design that compels the encoder to produce a token sequence with both semantic richness and forward-dependency. Furthermore, by incorporating prefix tokens and employing a two-stage tokenizer training process to enhance reconstruction performance, AliTok achieves high fidelity and predictability simultaneously. Building upon AliTok, a standard decoder-only autoregressive model with just 177M parameters achieves a gFID of 1.44 and an IS of 319.5 on ImageNet-256. Scaling to 662M, our model reaches a gFID of **1.28**, surpassing the SOTA diffusion method with **10×** faster sampling. On ImageNet-512, our 318M model also achieves a SOTA gFID of **1.39**. Code and weights at AliTok.

## 1 INTRODUCTION

Autoregressive models, particularly GPT-style decoder-only transformers (Achiam et al., 2023; Touvron et al., 2023a;b), have achieved revolutionary success in natural language processing owing to their simple yet scalable next-token prediction paradigm. Inspired by this triumph, the research community is actively exploring the application of this powerful paradigm to image generation (Team et al., 2025; Han et al., 2024; Wei et al., 2025) by sequentially predicting a stream of tokens, showcasing a promising path for multi-modal unification (Team, 2024; Wu et al., 2025; Chen et al., 2025).

---

[*]Equal contribution.    [†] Corresponding author.

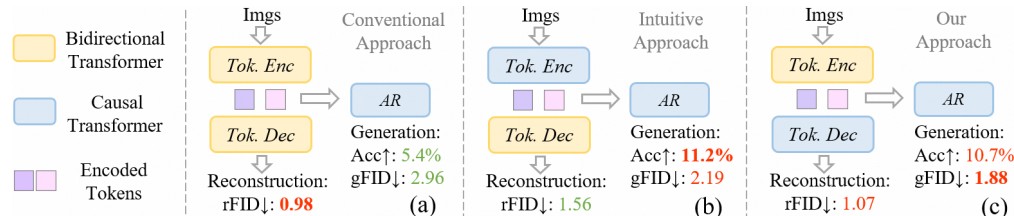

Figure 2: **Reconstruction** *vs* **generation** with different transformer-based tokenizers. Images are compressed into raster-scan order 1D sequences by the tokenizers. AR are standard decoder-only autoregressive models. green for poor results and red for good results. The best results are **bolded**. Tok. : Tokenizer. Acc: Training accuracy. Fair setup with matched parameter counts and computational loads. See Appendix A.2 for details.

However, as demonstrated in previous literature (Pang et al., 2024; Li et al., 2024b), the inherent bi-directional property of visual sequences makes it difficult for the raster-scan decoder-only autoregressive model (*e.g.*, LlamaGen (Sun et al., 2024)) to achieve excellent performance, since the unidirectional transformer is hard to model bi-directional sequences. Consequently, recent works have shifted focus towards paradigms like masked autoregressive (Li et al., 2024b; Chang et al., 2022; Yu et al., 2024c; Fan et al., 2024) and next-scale prediction (Tang et al., 2024; Tian et al., 2024; Han et al., 2024), which employ bidirectional attention in autoregressive models and demonstrate a superior choice for visual modeling. However, these approaches complicate visual generation and diverge from the traditional autoregressive paradigm, increasing the challenges for multi-modal unification. *In contrast to existing works that modify the model to fit the data properties, we propose an alternative perspective: Can we instead instill a forward-dependency into the token sequence, to align with the powerful simplicity of decoder-only AR models?*

Unlike natural language, which is inherently compact and allows for one-to-one mapping between words and indices with a non-parametric tokenizer, images are high-dimensional and redundant, requiring a learnable tokenizer for effective compression. In pursuit of maximal reconstruction fidelity, conventional tokenizer implicitly incentivizes global collaborative encoding among all tokens to efficiently eliminate redundancy. This means the representation of a token becomes functionally dependent on its non-causal context, particularly on subsequent tokens in the raster-scan order. This bidirectional dependency of representations creates a fundamental conflict with the strictly unidirectional predictive paradigm of autoregressive models. Since the target token depends on unseen future content, the conditional probability distribution learned by the AR model exhibits extremely high uncertainty, rendering the learning task exceedingly complex and limiting its generative capability.

To empirically validate this analysis, we construct an intuitive method where the encoder is forced to adopt a causal structure, strictly forbidding the preceding image patches access to future information and thus ensuring a purely unidirectional dependency. As shown in Fig. 2(b), the results are instructive: the training accuracy of AR model skyrockets from 5.4% to 11.2%, demonstrating that the predictability of sequence is substantially improved. However, this loss of global receptive field catastrophically degrades reconstruction quality (rFID: 0.98→1.56). A core challenge therefore emerges: Can we preserve the efficient compression of a bidirectional encoder while ensuring its output token sequence possesses the high predictability required by AR models?

To address this dilemma, we propose AliTok, whose core insight is to decouple the process of global semantic construction from the causal constraints of sequence. Specifically, AliTok leverages the global receptive field of a bidirectional encoder to build semantically rich representations, but critically, couples it with a causal decoder that acts as a powerful implicit regularizer. During reconstruction, the visibility of encoded tokens is strictly confined to the causal context along a raster-scan order. This architectural constraint, in turn, forces the encoder to suppress non-causal dependencies from its representations, thereby ensuring that the relevant contextual information required for reconstruction is efficiently organized within each token's causal history. Ultimately, this process yields a token sequence that is both semantically rich and highly predictable. Experiments in Fig. 2(c) demonstrate that this approach achieves high reconstruction fidelity (rFID: 1.07) while dramatically enhancing generative performance (gFID: 2.96→1.88).

Building upon this core design, we introduce several key implementation details to address its practical challenges and maximize performance. While the causal, raster-scan decoding is effective at enforcing forward-dependency, it leads to poor reconstruction of the initial tokens (*i.e.*, the first row

of the image) due to a lack of preceding context. To address this problem, we introduce prefix tokens specific to the first row, guided by a dedicated auxiliary loss to provide the necessary contextual priors for compensation. Finally, to further enhance reconstruction quality without compromising its generation-friendliness, we introduce a two-stage tokenizer training strategy. The second stage retrains a powerful bidirectional decoder over the frozen encoder and codebook, significantly boosting visual continuity and detail consistency.

To validate the effectiveness of AliTok, we employ a standard decoder-only autoregressive model as the generative model and conduct experiments. The results, illustrated in Fig. 3, demonstrate a remarkable leap in performance and efficiency. Even with a 318M parameter model, our method already surpasses all methods on gFID (w/o cfg), including state-of-the-art diffusion model LightningDiT (Yao et al., 2025). Upon increasing the parameter count

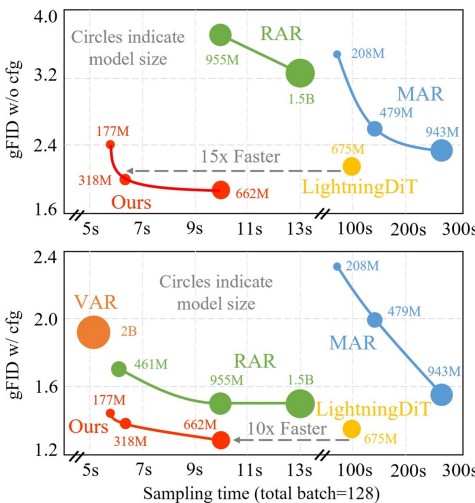

Figure 3: **Sampling time and gFID** (w/o cfg and w/ cfg). Sampling time is evaluated on an A800.

to 662M, we achieve a better gFID (w/ cfg) than LightningDiT (1.28 $vs$ 1.35), while providing a $10\times$ faster sampling speed. In summary, the contributions of this paper include:

1. We reveal a key factor restricting autoregressive model efficacy: conventional tokenizers tend to establish bidirectional dependencies within encoded tokens, which fundamentally conflicts with the unidirectional nature of autoregressive models.

2. We propose a simple yet effective design for an image tokenizer that enables the encoded tokens to be more easily modeled by autoregressive models while ensuring high reconstruction fidelity.

3. Based on the proposed tokenizer, a standard decoder-only autoregressive model beats state-of-the-art diffusion models on ImageNet benchmark.

## 2 RELATED WORK

**Autoregressive visual generation.** Inspired by GPT-style language models (Achiam et al., 2023; Touvron et al., 2023a;b), early visual autoregressive methods (Van Den Oord et al., 2017; Lee et al., 2022) predict image tokens along the raster-scan order. However, this strict unidirectional paradigm struggles to effectively model the bidirectional spatial dependencies inherent in visual sequence, thereby limiting its performance ceiling. To resolve this fundamental conflict, research has pivoted to re-engineering the models or generation paradigms for stronger bidirectional context awareness. One prominent approach is masked autoregressive modeling, including MaskGIT (Chang et al., 2022) and MAR (Li et al., 2024b), which leverages bidirectional attention for multi-round parallel generation. Another category explores hierarchical or next-scale prediction, such as VAR (Tian et al., 2024), which proceeds autoregressively across scales yet bidirectionally within each scale. Furthermore, other efforts enhance global perception by maximizing all-directional generation probabilities (RAR, Yu et al. (2024b)) or adopting random generation orders (RandAR, Pang et al. (2024)). While effective, these approaches compromise the simplicity of the classical autoregressive paradigm by introducing complex generation mechanisms or training objectives. Unlike these strategies that adapt models to data, this paper focuses on reshaping the data itself. We propose a novel tokenizer that instills a causal dependency structure into the token sequence during the encoding stage, thereby enabling a standard decoder-only autoregressive model to unleash its full potential.

**Image tokenizer**, a type of Variational Autoencoder (VAE) (Kingma et al., 2013), plays an important role in visual generation domain (Wan et al., 2025; Wu et al., 2024; Wei et al., 2024; Tang et al., 2025; Peng et al., 2025). By mapping high-dimensional image pixels to a low-dimensional latent space, the image tokenizer significantly enhances the training efficiency of generative models. Among them, VQ-VAE (Van Den Oord et al., 2017) utilizes quantization techniques to discretize continuous latent features for autoregressive image generation. Subsequently, VQ-GAN (Esser et al.,

2021) substantially improves visual fidelity by incorporating an adversarial loss. MagViT-v2 (Yu et al., 2024a) and FSQ (Mentzer et al., 2023) focus on algorithmic improvements to efficiently scale the codebook size. Besides, TiTok (Yu et al., 2024c) proposes a transformer-based tokenizer, encoding 2D image patches into a 1D sequence. FlexTok (Bachmann et al., 2025) explores multi-granularity semantic encoding, ranging from coarse to fine-grained. Despite numerous efforts, there has been limited exploration of how to design a tokenizer that simultaneously achieves high-quality reconstruction while being more conducive to subsequent autoregressive generation. Bridging this critical gap is the core goal of our work.

## 3 METHOD

### 3.1 PRELIMINARIES

**The modeling assumption of the autoregressive model** is to factorize the joint probability distribution of a sequence $x = [x_1, \ldots, x_T]$ into a product of unidirectional conditional probabilities:

$$p_\theta(x) = \prod_{i=1}^{T} p_\theta(x_i \mid x_1, \ldots, x_{i-1}), \tag{1}$$

where $p_\theta$ is the parameterized autoregressive model. Consequently, the learning efficiency of an autoregressive model fundamentally depends on whether its modeling target, the token sequence $x$, possesses a strong unidirectional dependency structure. When the critical information required to predict $x_i$ is predominantly contained within its preceding context $[x_1, \ldots, x_{i-1}]$, the learning task for the model becomes significantly more well-defined and efficient.

**Image tokenizer.** The inherent spatial continuity in images leads to redundancy not only within individual patches but also extensively among adjacent ones. To achieve effective compression, multiple tokens must collaborate to eliminate overall redundancy and construct a compact representation, which inevitably creates strong bidirectional representational dependencies among encoded tokens. This directly conflicts with the autoregressive learning paradigm: the model aims to learn the conditional distribution $p(x_i \mid x_{<i})$, while the ground-truth token $x_i$ has a representation that implicitly depends on the future context $x_{>i}$. Consequently, the target conditional distribution $p(x_i \mid x_{<i})$ becomes a marginalization over all possible unseen futures $x_{>i}$. Such a distribution is inherently high-entropy, severely limiting the model's convergence and generation quality.

### 3.2 ALITOK TOKENIZER

To resolve the dilemma posed in Introduction (Sec. 1), we introduce AliTok, a novel tokenizer. Its core insight is to employ a causal decoder to constrain the training of a bidirectional encoder, compelling it to produce a highly predictable token sequence while retaining its efficient compression capabilities. Furthermore, to address the issue of insufficient initial context arising from the causal constraint, we introduce extra prefix tokens for contextual priming. Finally, we mitigate the core trade-off by adopting a two-stage tokenizer training process: the first stage learns a generation-friendly encoder, while the second stage retrains a separate, high-fidelity bidirectional decoder.

**Definition.** As depicted in Fig. 4, the architecture of AliTok is built upon a vanilla vision transformer. An input image $I \in \mathbb{R}^{h \times w \times 3}$ is first processed into a sequence of patch tokens $P \in \mathbb{R}^{(H \times W) \times D}$. Here, $H = h/f, W = w/f$, $D$ is the number of channels, and $f$ is patch size (set to 16). Simultaneously, we introduce $K + H \times W$ latent tokens in the input space (with $K = W = 16, H \times W = 256$) as information carriers, with the first $K$ tokens serving as prefix tokens, and the remaining $H \times W$ latent tokens corresponding to $H \times W$ patch tokens. After concatenating the latent tokens with the patch tokens, the sequence is fed into the encoder (Enc) for tokenization, which compresses the information of the image patches into 1D latent tokens. Subsequently, the patch tokens are discarded, and we retain only the encoded latent tokens, denoted as $Z \in \mathbb{R}^{(K + H \times W) \times D}$. Finally, these tokens $Z$ undergo vector quantization (Quant) (Van Den Oord et al., 2017) (see Appendix A.3 for details) and pass through the decoder (Dec) to reconstruct the image patch sequence $\{\hat{p}_n\}_{n=1}^{K + H \times W}$.

**Causal decoder.** The core of our method lies in the first stage of training, which aims to produce a generation-friendly encoder and codebook. To achieve this, we introduce a causal decoder as a

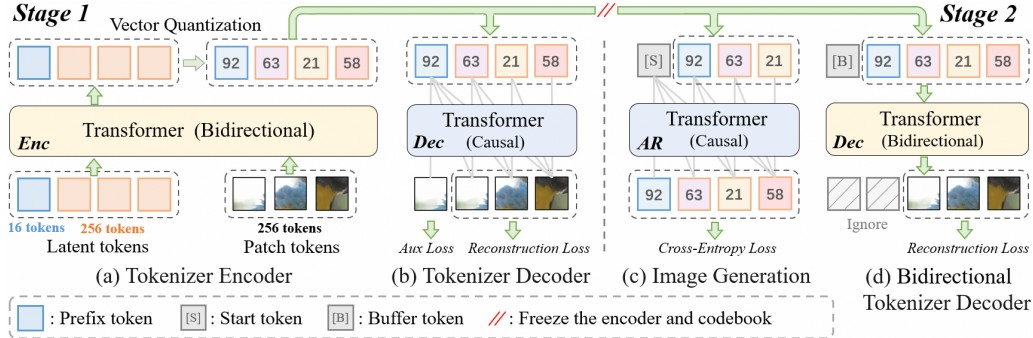

Figure 4: **Two-stage training process of the proposed AliTok.** Stage 1: Training an image tokenizer with a causal decoder. Stage 2: Freezing the encoder and codebook of the tokenizer, training the autoregressive model and retraining a bidirectional tokenizer decoder.

critical architectural constraint. Unlike a conventional bidirectional decoder, the causal decoder's visibility of the encoded tokens is strictly confined to a causal, raster-scan order during reconstruction. Specifically, the reconstruction of the $i$-th image patch $\hat{p}_i$ is conditioned only on its causal context. This process is formally defined as:

$$\{\hat{p}_k\}_{k=1}^i = \text{Dec}_{\text{causal}}(\{\text{Quant}(z_k)\}_{k=1}^i). \tag{2}$$

This architectural constraint acts as a powerful implicit regularizer. It compels the bidirectional encoder to alter its encoding behavior in order to minimize reconstruction loss under this limited receptive field, ultimately learning an autoregressively-structured representation, where the contextual information required to reconstruct patch $p_i$ is strategically organized within the causal sequence $z_{1...i}$. This learned strategy establishes a forced alignment between the token dependency structure and the autoregressive generation process (Fig. 4(b) and (c)). As a result, the next-token prediction task becomes substantially more well-defined for the AR model, enabling a more stable and effective training process that directly leads to higher generation quality.

The reconstruction loss $L_{recon}$ for the first stage, follows standard practices (Yu et al., 2024c), combining a mean squared error (MSE) loss $L_{mse}$, perceptual loss $L_{perc}$, quantization loss (Zheng & Vedaldi, 2023) $L_{quant}$, and adversarial loss $L_{adv}$:

$$L_{recon} = L_{mse} + L_{perc} + L_{quant} + \lambda L_{adv}, \tag{3}$$

where $\lambda$ is set to 0.1. For adversarial loss, we employ the GAN in Open-MagViT2 (Luo et al., 2024).

**Prefix token.** However, a causal decoder forces latent tokens to only reference features from preceding tokens, leading to poor reconstruction of the first row (16×256 pixels) of the image. To solve this problem, an intuitive solution is to add an extra row to the training image, *i.e.*, using images with 272×256 resolution for training, and crop out the unsatisfactory first row in the final generated result. But this simple approach may result in generating images with incomplete objects, as information in the topmost row is lost during cropping.

To provide contextual priors for the problematic first row, we introduce 16 prefix tokens as supplementary aids, each dedicated to a patch in the first row. These tokens are optimized via a specialized auxiliary loss, $L_{aux}$, which incorporates both MSE and perceptual losses. Since the perceptual loss requires a full image for accurate assessment, we form a complete image by concatenating the first row, reconstructed from the prefix tokens, with the remaining 15 rows, reconstructed from the last 240 latent tokens. Critically, we detach the gradient from the latter component before concatenation. This design allows the perceptual network to evaluate a spatially coherent image while ensuring the optimization signal is backpropagated only to the reconstruction result of prefix tokens.

**Retraining a bidirectional decoder.** In the second stage, we freeze the encoder and codebook of the tokenizer and retrain a bidirectional decoder to improve detail consistency. As shown in Fig. 4(d), in addition to converting the attention mechanism in the decoder, we integrate 64 buffer tokens (Li et al., 2024b) to enhance modeling capabilities by increasing computational load. Meanwhile, the loss previously imposed on prefix tokens is removed to allow the decoder to focus on reconstruction quality. This two-stage training strategy enables our AliTok tokenizer to not only produce generation-friendly encoded tokens, but also maintain good reconstruction performance.

Table 1: **ImageNet 256×256 conditional generation.** "Diff.": Diffusion. "Mask.": Masked transformer models. Pre.: Precision. Rec.: Recall. RAR does not report results w/o cfg. Thus, we test it using the weights provided in the original paper, adjusting the temperature at intervals of 0.01 to select the best gFID w/o cfg.

| Type | Generator | Training epochs | #Para. | w/o cfg | | w/ cfg | | | |
|---|---|---|---|---|---|---|---|---|---|
| | | | | gFID↓ | IS↑ | gFID↓ | IS↑ | Pre.↑ | Rec.↑ |
| Diff. | SiT-XL (Ma et al., 2024) | 800 | 675M | 8.61 | 131.7 | 2.06 | 270.3 | 0.82 | 0.59 |
| | REPA (Yu et al., 2024d) | 800 | 675M | 5.90 | 157.8 | 1.42 | 305.7 | 0.80 | 0.65 |
| | LightningDiT (Yao et al., 2025) | 800 | 675M | 2.17 | 205.6 | 1.35 | 295.3 | 0.79 | 0.65 |
| VAR | VAR-d24 (Tian et al., 2024) | 350 | 1.0B | – | – | 2.09 | 312.9 | 0.82 | 0.59 |
| | VAR-d30 (Tian et al., 2024) | 350 | 2.0B | – | – | 1.92 | 323.1 | 0.82 | 0.59 |
| Mask. | MaskGIT (Chang et al., 2022) | 300 | 227M | 6.18 | 182.1 | – | – | – | – |
| | MAGVIT-v2 (Yu et al., 2024a) | 1080 | 307M | 3.65 | 200.5 | 1.78 | 319.4 | – | – |
| | TiTok-S-128 (Yu et al., 2024c) | 800 | 287M | 4.44 | 168.2 | 1.97 | 281.8 | – | – |
| MAR | MAR-B (Li et al., 2024b) | 800 | 208M | 3.48 | 192.4 | 2.31 | 281.7 | 0.82 | 0.57 |
| | MAR-L (Li et al., 2024b) | 800 | 479M | 2.60 | 221.4 | 1.98 | 290.3 | 0.81 | 0.60 |
| | MAR-H (Li et al., 2024b) | 800 | 943M | 2.35 | 227.8 | 1.55 | 303.7 | 0.81 | 0.62 |
| Causal AR | LlamaGen-XL (Sun et al., 2024) | 300 | 775M | – | – | 2.62 | 244.1 | 0.80 | 0.57 |
| | LlamaGen-XXL (Sun et al., 2024) | 300 | 1.4B | – | – | 2.34 | 253.9 | 0.80 | 0.59 |
| | LlamaGen-3B (Sun et al., 2024) | 300 | 3B | 9.38 | 112.9 | 2.18 | 263.3 | 0.81 | 0.59 |
| | RAR-B (Yu et al., 2024b) | 400 | 261M | 7.12 | 124.8 | 1.95 | 290.5 | 0.82 | 0.58 |
| | RAR-L (Yu et al., 2024b) | 400 | 461M | 5.39 | 149.1 | 1.70 | 299.5 | 0.81 | 0.60 |
| | RAR-XL (Yu et al., 2024b) | 400 | 955M | 3.72 | 179.9 | 1.50 | 306.9 | 0.80 | 0.62 |
| | RAR-XXL (Yu et al., 2024b) | 400 | 1.5B | 3.26 | 193.3 | 1.48 | 326.0 | 0.80 | 0.63 |
| Causal AR | AliTok-B | 800 | 177M | 2.40 | 177.1 | 1.44 | 319.5 | 0.77 | 0.65 |
| | AliTok-L | 800 | 318M | 1.98 | 200.8 | 1.38 | **326.2** | 0.78 | 0.65 |
| | AliTok-XL | 400 | 662M | **1.88** | **228.4** | **1.28** | 306.3 | 0.79 | 0.65 |

## 3.3 AUTOREGRESSIVE MODEL

Our decoder-only autoregressive architecture follows the standard design in LlamaGen (Sun et al., 2024), which uses RMSNorm (Zhang & Sennrich, 2019) for pre-normalization and applies 2D rotary positional embeddings (RoPE) (Su et al., 2024) at each layer. Building upon LlamaGen, we make only a few modifications. First, since our method has 16 extra prefix tokens and needs to model 272 tokens, 2D RoPE cannot be used directly. Therefore, we use 1D RoPE for the prefix tokens and 2D RoPE for the remaining 256 tokens. Secondly, we introduce the QK-Norm operation (Team, 2024) in the attention module to stabilize the large-scale model training, aligning with recent multi-modal autoregressive models (Xie et al., 2024; Ma et al., 2025; Team, 2024). We do not conduct extensive exploration of autoregressive models and only use the conventional architecture to verify the effectiveness of the proposed tokenizer.

## 4 EXPERIMENTS

### 4.1 EXPERIMENTAL SETUP

**Models and evaluations.** The design of the proposed tokenizer is based on TA-TiTok (Kim et al., 2025), utilizing ViT-B (Yu et al., 2024c) for the encoder and ViT-L for the decoder. We set the vocabulary size to 4096 and adopt the online feature clustering method (Zheng & Vedaldi, 2023) to ensure that the codebook utilization is 100%. For simplicity, we call the autoregressive models trained using our AliTok tokenizer as AliTok-B/L/XL, comprising three different sizes, more details are provided in Appendix A.4. Following common practice, FID (including rFID and gFID) (Heusel et al., 2017), IS (Salimans et al., 2016), Precision, and Recall are adopted as metrics. We generate 50,000 images to test the gFID and adopt the evaluation code from RAR (Yu et al., 2024b). For rFID, we use the ImageNet-1K val set for evaluation, consistent with other methods (Sun et al., 2024; Yu et al., 2024c; Yao et al., 2025). During the test, KV-cache is employed to enhance sampling speed.

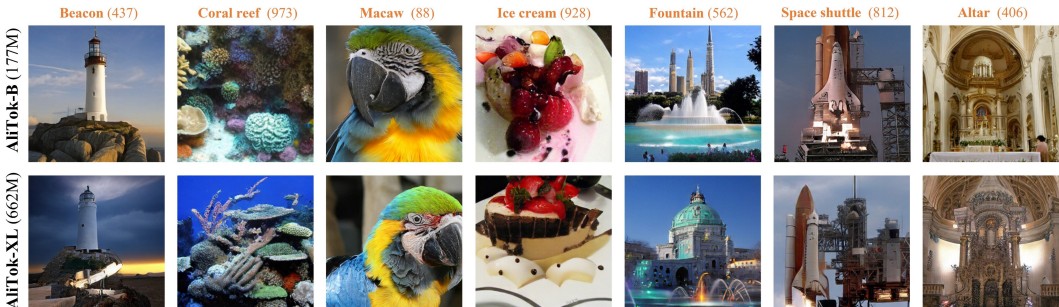

Figure 5: **256×256 samples generated** by our models of different sizes.

**Dataset and training details.** We train our image tokenizer from scratch on ImageNet-1K (Russakovsky et al., 2015). The tokenizer is trained for 600K steps in the first stage and 300K steps in the second stage on 32 A800-80G GPUs. For the autoregressive model training, we employ tencrop (Szegedy et al., 2015) for data augmentation and cache (Li et al., 2024b) the encoded results of the tokenizer to reduce training time. We train the base model and large model for 800 epochs and the XL model for 400 epochs, with a batch size of 2048 and a learning rate of 4e-4. For the first 100 epochs, the learning rate is linearly warmed up, then decayed using a cosine decay schedule down to 1e-5, following RAR (Yu et al., 2024b). During training, class conditioning is randomly dropped with a probability of 0.1 to support the use of classifier-free guidance (cfg) (Ho & Salimans, 2022).

## 4.2 MAIN RESULTS

**Generation comparison.** We report a comprehensive comparison with state-of-the-art methods on the ImageNet-1K 256×256 benchmark in Table 1. Utilizing our AliTok image tokenizer, even a standard autoregressive model demonstrates exceptional performance. Specifically, our AliTok-B model achieves a gFID (w/ cfg) of 1.44, using less than 6% of the parameter count compared to its raster-order counterpart LlamaGen-3B (Sun et al., 2024), which achieves a gFID of 2.18 with a 3B model. Furthermore, our AliTok-L model, with a parameter count of 318M, attains a gFID (w/ cfg) of 1.38, eclipsing all existing autoregressive methods and outperforming the state-of-the-art autoregressive model RAR-XXL (1.5B parameters) (Yu et al., 2024b) in both IS and gFID. Upon increasing the parameter count to 662M, our AliTok-XL beats all competing methods, particularly in gFID w/o cfg, markedly outperforming the previous best method LightningDiT (Yao et al., 2025) (1.88 *vs* 2.17). When using the cfg, our AliTok-XL achieves a gFID of 1.28, surpassing LightningDiT (1.35) while also achieving a superior IS. To the best of our knowledge, this represents the first time a standard autoregressive model beats state-of-the-art diffusion models.

**Sampling efficiency comparison.** Beyond its excellent generation quality, AliTok also demonstrates a significant advantage in sampling efficiency benefiting from the use of KV-cache. As shown in Table 2, with similar gFID scores, AliTok-L improves throughput by 33.7× and 2.0× compared to MAR-H (Li et al., 2024b) and RAR-XXL (Yu et al., 2024b), respectively. Compared to the state-of-the-art diffusion model LightningDiT (Yao et al., 2025), our AliTok-XL requires less than 10% of the time needed by LightningDiT to produce an image, establishing a substantial improvement in image generation efficiency.

Table 2: **Sampling speed comparison.** Throughput (images/sec) measured on an A800 GPU (FP32, batch size=64), using the original codebases for all methods.

| Method | Type | #Para. | gFID↓ | images/sec↑ |
|---|---|---|---|---|
| VAR-d30 | VAR | 2.0B | 1.92 | 12.3 |
| AliTok-B (ours) | AR | 177M | 1.44 | 10.9 |
| MAR-H | MAR | 943M | 1.55 | 0.3 |
| RAR-XXL | AR | 1.5B | 1.48 | 5.0 |
| AliTok-L (ours) | AR | 318M | 1.38 | 10.1 |
| LightningDiT | Diff. | 675M | 1.35 | 0.6 |
| AliTok-XL (ours) | AR | 662M | 1.28 | 6.3 |

**Generation visualization.** We further show the generation results under different model sizes in Fig. 5. It can be observed that even the smallest model is capable of producing high-quality visual results and reasonably generating complex structures, such as the coral reef and altar. As the model size increases, the generated images exhibit enhanced detail and more intricate structures, illustrated by the realistic texture of the feathers of the parrot, the fine-grained structure of the space shuttle

Figure 6: **Training curves**. (a) training loss (b) training error rate (%) (c) gFID scores w/o cfg (d) gFID scores w/ cfg varies with training steps and model parameters. 250,000 training steps corresponds to 400 epochs.

Table 3: **Ablation studies of main components** on AliTok-Base model. Experiment (A) employs a bidirectional Transformer as the baseline. Subsequent experiments progressively incorporate a causal tokenizer decoder (Causal Dec), prefix tokens (Prefix), auxiliary loss ($L_{aux}$), and two-stage tokenizer training strategy (Two-stage). Key metrics including AR training loss (Loss), AR training accuracy (Acc.), generation gFID, and reconstruction rFID are reported to show the impact of each component.

| Training Setting | | Tokenizer Setting | | | | AR Training | | Evaluation | | |
|---|---|---|---|---|---|---|---|---|---|---|
| | | Causal Dec | Prefix | $L_{aux}$ | Two-stage | Loss↓ | Acc.↑ | gFID↓ | IS↑ | rFID↓ |
| Tokenizer trained for 300K steps & AR for 200 epochs | **(A)** | | | | | 5.75 | 5.4% | 2.96 | 270.6 | 0.98 |
| | **(B)** | ✓ | | | | 5.18 | 10.7% | 1.88 | 285.6 | 1.07 |
| | **(C)** | ✓ | ✓ | | | 5.20 | 9.7% | 1.86 | 283.5 | 1.01 |
| | **(D)** | ✓ | ✓ | ✓ | | 5.06 | 10.2% | 1.82 | 286.5 | 1.02 |
| Best rFID & AR for 800 epochs | **(E)** | ✓ | ✓ | ✓ | | **4.98** | **10.5%** | 1.47 | 316.3 | 0.91 |
| | **(F)** | ✓ | ✓ | ✓ | ✓ | **4.98** | **10.5%** | **1.44** | **319.5** | **0.86** |

and the altar, demonstrating a superior level of realism in image quality. More results generated by the different model sizes are provided in the Appendix (Fig. 14, 15, 16, 17, and 18).

**Training curves.** As shown in Fig. 6, we explore the training behavior of the proposed AliTok. With the increase in model parameters and training steps, both the training loss and training error rate decrease significantly and steadily. Additionally, larger models achieve lower loss and superior gFID scores with fewer training steps. Experiments verify that our generative model effectively inherits the scaling ability of autoregressive models. However, the gFID curves indicate that after 400 training epochs, neither our AliTok-B model nor AliTok-L model shows clear signs of convergence. Therefore, we extend the training duration for these two models to 800 epochs to further improve the performance. More analysis is provided in Appendix A.4.

**Reconstruction performance.** We report the reconstruction performance of various tokenizers (Yu et al., 2024c; Zheng & Vedaldi, 2023; Xiong et al., 2025) in Table 4. By introducing a powerful bidirectional decoder in the second stage, AliTok significantly improves the rFID to 0.86. This result remains competitive with GigaTok-B-L (rFID: 0.81), particularly given that the latter relies on a substantially larger vocabulary size (16384 $vs$ 4096). Experiments demonstrate that our proposed AliTok successfully achieves an ideal balance between the goals of generation-friendly and high-fidelity through its ingenious decoupling design.

Table 4: **Reconstruction quality.** Size: vocabulary size.

| Tokenizer | Tokens | Size | rFID↓ |
|---|---|---|---|
| CVQ-VAE | 256 | 1024 | 1.57 |
| TiTok-B-64 | 64 | 4096 | 1.70 |
| GigaTok-B-L | 256 | 16384 | 0.81 |
| AliTok (Stage 1) | 272 | 4096 | 0.91 |
| AliTok (Stage 2) | 272 | 4096 | 0.86 |

### 4.3 ABLATION STUDIES

**Ablation studies of main components.** In Table 3, we ablate the main components of the proposed AliTok. First, introducing a causal decoder (B) dramatically improves predictability over the bidirectional baseline (A). This is evidenced by the AR model's training accuracy soaring from 5.4% to 10.7% and its gFID dropping from 2.96 to 1.88, confirming the critical role of aligning token dependencies with the model's causal nature. Next, we examine the effect of prefix tokens. Simply adding them (C) not only increases the modeling burden due to a longer sequence, but these unconstrained tokens also tend to learn global features. This makes the prediction task for the autoregressive model more difficult and fails to alleviate the poor first-row reconstruction (Fig. 7(e)). In contrast, when

| (a) Original image | (b) Baseline | (c) Baseline + Causal Enc | (d) Baseline + Causal Dec | (e) + Prefix Tokens | (f) + $L_{aux}$ | (g) + Two Stage |

Figure 7: **Reconstruction visualization comparison**. (a) Original image. (b, c) Comparison results. (d-g) Ablation study showing the progressive addition of our components. red boxes highlight noticeable artifacts.

guided by the auxiliary loss $L_{aux}$ (D), the prefix tokens are forced to represent the features of the first image row. This creates a more progressive and consistent learning process, as the AR model consistently predicts a token based on adjacent context, unifying the prediction task and leading to a rebound in training accuracy (10.2%) and a better gFID of 1.82. Finally, by reintroducing a bidirectional decoder (F) through a two-stage training process, AliTok further enhances reconstruction fidelity and improves generative quality.

**Ablation studies of reconstruction visualization.** We visualize the impact of each component on reconstruction quality in Fig. 7. First, adopting a causal encoder (c) introduces noticeable visual discontinuities and grid-like artifacts (highlighted by red boxes), as its unidirectional structure limits compression efficiency. In contrast, a causal decoder (d) yields generally natural reconstructions, but its sequential reconstruction process leads to poor quality in the first row, marked by severe blurring and distortion at the top of the bookshelf. To address this, we introduce prefix tokens with $L_{aux}$ (f) to provide the necessary contextual prior, leading to a significant restoration of detail in the first row. Finally, retraining a bidirectional decoder in the second stage (g) smooths patch boundaries and rectifies detail inaccuracies, such as the lack of clarity on the metal mesh. Through these sequential designs, our final reconstruction (g) becomes visually comparable to the baseline (b).

**Visualization of attention maps.** To demonstrate the intrinsic dependency structure within AliTok's latent tokens, we analyze the attention maps from the bidirectional decoders of both baseline model and AliTok (Stage 2) in Fig. 8. We first examine the mean local (3×3) attention pattern averaged across all tokens in Fig. 8(b). As expected, the baseline's attention is symmetrically distributed, indicating a balanced reliance on past and future context. In stark contrast, AliTok's mean attention map reveals a clear systemic shift towards the top-left, demonstrating a strong preference for causally preceding tokens even with full access to future ones. To demonstrate the generality of this behavior, Fig. 8(c) displays attention maps for several uniformly sampled tokens, which consistently exhibit this same causal bias. More examples are provided in the Appendix (Fig. 11). This striking asymmetry provides compelling visual evidence that our training strategy effectively regularizes the encoder. It compels the encoder to learn autoregressively-structured representations, where the necessary context for reconstruct-

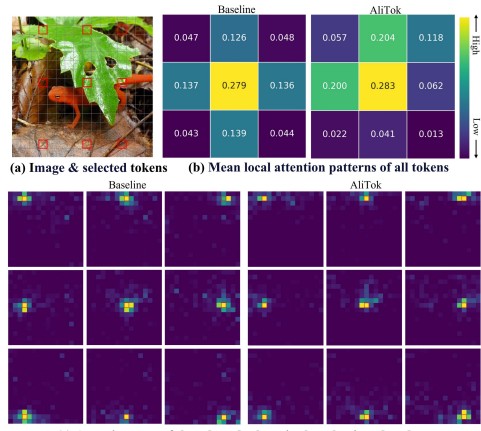

Figure 8: **Visualization of attention maps from bidirectional decoders.** (a) The input image and selected tokens (red boxes) for analysis. (b) The mean local (3×3) attention patterns across all tokens. (c) Attention maps for the selected tokens.

ing a patch is primarily packed into its causal tokens. This process instills a strong causal bias into the token sequence itself, shaping it to be inherently amenable to forward prediction. Such fundamental alignment unlocks the high performance of a standard autoregressive model.

**Scalability in high-resolution generation.** To verify the scalability of our method on the high-resolution image generation task, we trained AliTok on the ImageNet dataset at 512×512 resolution. For this experiment, both the tokenizer and autoregressive model are fine-tuned from pretrained weights at 256 resolution, where the embedding of tokenizer is linearly interpolated to match the shape. The generation results are shown in Table 5. Our AliTok-L achieves

Table 5: **512×512 generation quality.**

| Method | #params | gFID↓ | IS↑ |
|---|---|---|---|
| REPA | 675M | 2.08 | 274.6 |
| MAR-L | 481M | 1.73 | 279.9 |
| AliTok-L | 318M | **1.39** | **295.3** |

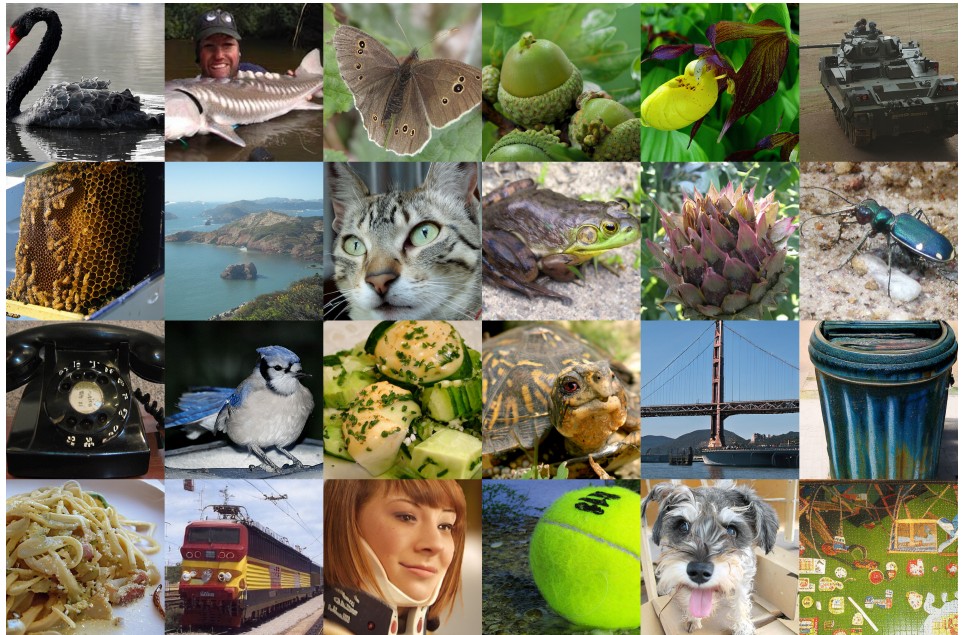

Figure 9: **512×512 samples** of class-conditional generation on ImageNet using our AliTok-L model (318M).

a gFID of 1.39 and an IS of 295.3, significantly surpassing MAR-L (Li et al., 2024b) and REPA (Yu et al., 2024d). The result highlights the strong scalability and robustness of our AliTok. Fig. 9 provides generated samples, visually confirming the high quality of the outputs.

**Scalability in codebook size.** To further validate the adaptability of the AliTok framework to larger discrete vocabularies, we conducted an experiment expanding the codebook size from 4096 to 16384. As shown in Table 6, the 16384 codebook achieves a lower reconstruction FID (from 0.86 to 0.81), indicating a higher upper bound for potential generation quality. On the generation side, after training our XL-sized model for 400 epochs, the generative performance with the 16384 codebook (gFID: 1.30) is slightly inferior to that of the 4096 codebook (gFID: 1.28). We attribute this temporary performance gap to the increased difficulty of the next-token prediction task within a 4× larger vocabulary space, which is clearly evidenced by the sharp drop in training accuracy and the rise in loss. Consequently, we posit that with extended training epochs or an increase in model parameters, the tokenizer with 16384 codebook size is expected to achieve better generation performance.

Table 6: **AliTok-XL reconstruction and generation performance at different codebook sizes.** AR models are trained for 400 epochs.

| Size | rFID↓ | gFID↓ | IS↑ | Loss↓ | Acc↑ |
|------|-------|-------|-------|-------|-------|
| 4096 | 0.86 | **1.28** | **306.3** | **4.67** | **12.2%** |
| 16384 | **0.81** | 1.30 | 304.9 | 6.42 | 4.3% |

## 5 CONCLUSION

This paper introduces AliTok, a novel tokenizer designed to mitigate the fundamental misalignment between conventional tokenizers and autoregressive models. By employing a causal decoder to constrain a bidirectional encoder, AliTok instills a significant causal bias into the image token sequence, making it inherently compatible with the autoregressive paradigm. This strategy enables a standard decoder-only generator to surpass state-of-the-art diffusion models in both generation quality and sampling efficiency. Our work confirms that when data and the model paradigm are properly aligned, the concise autoregressive approach remains a powerful pathway toward building efficient, high-performance generative models, paving the way for future multimodal unification.

### ACKNOWLEDGMENTS

This work is supported by the National Natural Science Foundation of China (NSFC) under Grants 62225207, 62436008, 62306295, and 62576328. The AI-driven experiments, simulations and model training were performed on the robotic AI-Scientist platform of Chinese Academy of Sciences.

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

# A    APPENDIX

## A.1    LIMITATION AND FUTURE WORK

Benefiting from the AliTok tokenizer, our 662M AR model reaches a significantly high training accuracy of 12.2%. In this context, the reconstruction performance of the discrete tokenizer becomes a critical bottleneck for the generative performance. To test the limiting value of gFID that can be achieved under this tokenizer, we randomly select 50 real images per class from the training set and calculate the gFID after reconstruction (differently from rFID evaluation), obtaining a FID of 1.15. This implies that even if the accuracy in the generation phase is close to 100% (which is not possible), the generation performance still cannot reach the upper limit of 1.15. This experimental result explains the phenomenon observed in Fig. 6, where the training loss and error rate of AliTok-XL continue to decrease steadily, yet the gFID metric struggles to achieve further improvement. While our experiments with a 16384-size codebook have demonstrated a higher ceiling for reconstruction quality (as shown in Table 5), they also revealed an increased learning challenge for the autoregressive model. This paper does not explore generative models larger than our XL variant, and we leave the comprehensive exploration, such as training larger-scale models or extending the training duration on the 16384 codebook to fully unlock its generative potential to future work.

## A.2    EXPERIMENTAL DETAILS IN INTRODUCTION

The experiments depicted in Fig. 2 of the main paper are conducted under a controlled setting to ensure a fair comparison. For all three configurations, the tokenizer is trained for 300K steps, and the corresponding AliTok-Base autoregressive model is trained for 200 epochs. The conventional approach serves as our baseline. It employs a standard architecture with a bidirectional Transformer for both the encoder and decoder. The tokenizer for our approach uses a bidirectional encoder but is constrained by a causal decoder. This configuration is identical to the model presented in Table 3(b) of our ablation study. For the intuitive approach using a causal encoder, we consider the following factors to design the tokenizer: Firstly, it is challenging to compress the information within image patches into latent tokens with causal logic. Secondly, for a causal encoder, adding more learnable tokens does not effectively increase computational load, as the direction of interaction is limited. Therefore, we remove the 256 latent tokens from the input space of the encoder and treat the encoded patch tokens as the encoding result. To ensure fairness, we relocate the 256 removed latent tokens to the input space of the bidirectional decoder as buffer tokens. Concurrently, to maintain the computational load of the encoder, we reverse the model sizes of the encoder and decoder by using a ViT-L model for the encoder and a ViT-B model for the decoder. In this way, we ensure that the computational effort of each tokenizer is similar and the design is reasonable. Finally, the codebook utilization of each tokenizer is 100% to avoid the impact on the training accuracy.

## A.3    METHOD DETAILS

**Vector quantization.** After the latent tokens pass through the encoder, a continuous encoded latent vector $z$ is produced. The nearest vector to $z$, based on Euclidean distance, is then queried from the discrete codebook $\mathcal{Z}$ to serve as the quantized result. This vector quantization process can be formalized by the following equation:

$$Quant(z) = \underset{z_i \in \mathcal{Z}}{\arg\min} \|z - z_i\|. \qquad (4)$$

**Hyperparameters.** In the first stage, we define the total loss $L_{total}$ as a straightforward sum of the reconstruction loss ($L_{rec}$) and the auxiliary loss ($L_{aux}$), treating them with equal importance:

$$L_{total} = L_{rec} + L_{aux} \qquad (5)$$

**Positional embedding in autoregressive models.** In autoregressive models, we employ 1D RoPE for the 16 prefix tokens, with positions ranging from 0 to 15. Following this, we use 2D RoPE for the remaining $16 \times 16$ tokens, with positions extending from 16 to 31.

Table 7: **Improvements and curves of training 800 epochs**. 500,000 training steps corresponds to 800 epochs. Acc.: final training accuracy. Since we employ cosine learning rate decay instead of a fixed learning rate, we train the models for 800 epochs from scratch rather than continuing training based on the weights from the 400 epochs.

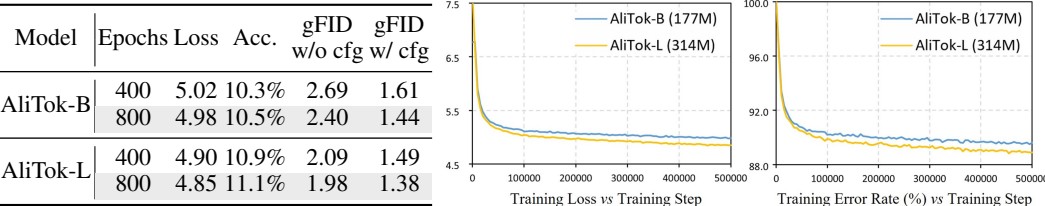

| Model | Epochs | Loss | Acc. | gFID w/o cfg | gFID w/ cfg |
|---|---|---|---|---|---|
| AliTok-B | 400 | 5.02 | 10.3% | 2.69 | 1.61 |
| | 800 | 4.98 | 10.5% | 2.40 | 1.44 |
| AliTok-L | 400 | 4.90 | 10.9% | 2.09 | 1.49 |
| | 800 | 4.85 | 11.1% | 1.98 | 1.38 |

Training Loss *vs* Training Step      Training Error Rate (%) *vs* Training Step

## A.4 EXPERIMENTS

**Model configurations.** For simplicity, we call the autoregressive models trained using our AliTok tokenizer as AliTok-B/L/XL, comprising three different sizes. For all autoregressive models, we set the number of attention heads to 16. Specifically, AliTok-B has a depth of 24 and a width of 768, AliTok-L has a depth of 24 and a width of 1024, and AliTok-XL has a depth of 32 and a width of 1280, following the RAR (Yu et al., 2024b) configuration.

**Improvements of training 800 epochs.** In Table 7, we show the enhancements achieved by increasing the training epochs from 400 to 800, which significantly reduces gFID scores both w/o cfg and w/ cfg. We further plot the training curves on the right side of Table 7. It is evident that the training loss and error rate continue to decrease linearly without convergence, suggesting that extending the number of training epochs may yield further improvements. However, since we use cosine learning rate decay rather than a fixed learning rate, additional epochs require training from scratch. Consequently, we do not explore the improvements that might result from extending the training period.

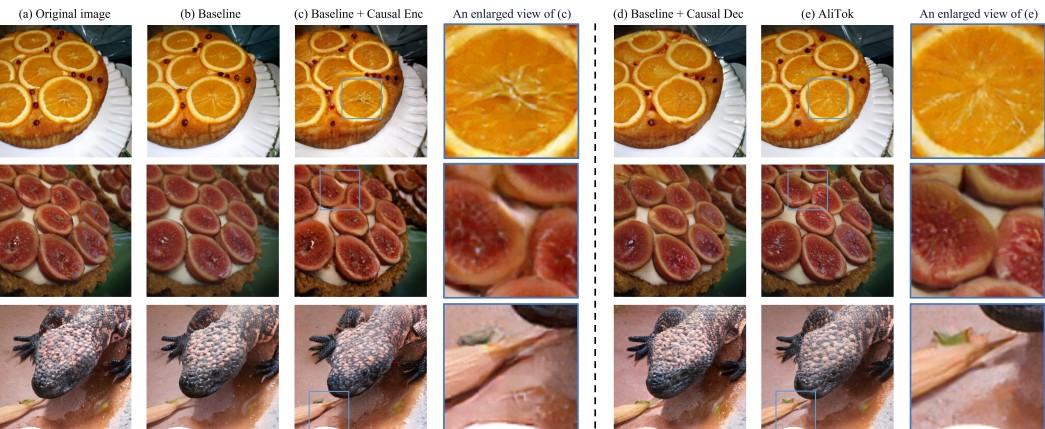

Figure 10: **More reconstruction result comparisons**. The same regions in (c) and (e) are enlarged for detailed comparison.

**More reconstruction results.** Fig. 10 presents additional comparisons of reconstruction results, with enlarged views of the detailed regions from both the intuitive approach (baseline + causal encoder) and our AliTok approach. From the enlarged region, it is obvious that the causal encoder causes texture mismatches between different reconstructed patches with discontinuities and detail deviations. Notably, significant grid artifacts persist even after the usage of adversarial loss. In contrast, our AliTok method achieves continuous and realistic reconstruction results. Furthermore, as seen in Fig. 10(d), solely applying the causal decoder on the baseline leads to color deviation and blurriness in the first row of the reconstructed images due to poor reconstruction quality, especially in the first image patch. In certain scenes, like the edge of the plate in the first row, unnatural transitions may also be present. However, after implementing our proposed series of improvements,

Table 8: Comparison of training configurations and model specifications.

| Tokenizer | TiTok-B | TiTok-L | GigaTok-B-L | AliTok (Stage 1) | AliTok (Stage 2) |
|---|---|---|---|---|---|
| Codebook size | 4096 | 4096 | 16384 | 4096 | 4096 |
| Codebook dimension | 12 | 12 | 8 | 32 | 32 |
| Training epochs | 200 | 200 | 200 | 120 | 60 |
| #Param | 204.8M | 641.1M | 621.6M | 389.8M | 390.0M |
| Total batch size | | 256 | | | 256 |
| Base learning rate | | 1e-4 | | | 1e-4 |
| Minimum learning rate | | 1e-5 | | | 1e-5 |

Table 9: **Hyperparameters for sampling** include without and with cfg.

| Model | Temperature | Scaler Power | Guidance Scale |
|---|---|---|---|
| AliTok-B | 0.95 | | w/o cfg |
| AliTok-L | 0.95 | | w/o cfg |
| AliTok-XL | 0.95 | | w/o cfg |
| AliTok-B | 1.00 | 1.3 | 11 |
| AliTok-L | 1.00 | 0.6 | 5 |
| AliTok-XL | 1.00 | 1.4 | 8 |

including prefix tokens, auxiliary loss, and a two-stage training strategy, this issue is significantly alleviated. Our AliTok (Fig. 10(e)) can ultimately achieve similar reconstruction quality and visual perception as the baseline approach (Fig. 10(b)).

**More generation results.** We present more generation results of our AliTok-XL model in Fig. 14. As can be seen, our method is capable of producing detailed textures and realistic image quality. In Fig. 15, Fig. 16, Fig. 17, Fig. 18, we present various images generated by AliTok-B and AliTok-XL across different categories. Even our smallest model can generate great visual outcomes.

**Tokenizer training specifications.** This subsection details the training configurations and model specifications for AliTok and other tokenizers, summarized in Table 8. As shown, our comparative experiments are conducted under fair conditions, with aligned batch sizes and learning rate schedules, ensuring the validity of our system-level performance analysis.

**Hyperparameters for sampling.** We list the sampling hyperparameter settings in Table 9. When cfg is not used, we employ a temperature hyperparameter. When cfg is used, we apply pow-cosine as the guidance schedule, following RAR (Yu et al., 2024b).

Table 10: System-level reconstruction performance comparison across different tokenizers.

| Tokenizer | Tokens | Size | PSNR↑ | SSIM↑ | LPIPS↓ | rFID↓ | #Param |
|---|---|---|---|---|---|---|---|
| TiTok-L-32 | 32 | 4096 | 16.15 | 0.5098 | 0.3232 | 2.21 | 641.1M |
| TiTok-B-64 | 64 | 4096 | 17.39 | 0.5491 | 0.2539 | 1.70 | 204.8M |
| AliTok (Stage 1) | 272 | 4096 | 20.66 | 0.6571 | 0.1659 | 0.91 | 389.8M |
| AliTok (Stage 2) | 272 | 4096 | **20.97** | **0.6610** | **0.1573** | **0.86** | 390.0M |
| GigaTok-B-L | 256 | 16384 | 21.24 | **0.6873** | **0.1284** | 0.81 | 621.6M |
| AliTok (Stage 1) | 272 | 16384 | 21.08 | 0.6805 | 0.1367 | 0.84 | 390.2M |
| AliTok (Stage 2) | 272 | 16384 | **21.30** | 0.6851 | 0.1303 | **0.81** | 390.4M |

**System-level comparison of reconstruction performance.** To comprehensively evaluate the performance of our proposed AliTok on the image reconstruction task, we detail the performance of AliTok against other tokenizers across several standard reconstruction metrics, including Peak Signal-to-Noise Ratio (PSNR), Structural Similarity Index Measure (SSIM), Learned Perceptual Image Patch Similarity (LPIPS), and reconstruction FID (rFID).

The results in Table 10 highlight AliTok's competitiveness. In the comparison with GigaTok-B-L using a 16384 codebook, our method achieves on-par reconstruction fidelity, matching its rFID of

0.81, while using approximately 37% fewer parameters (390.4M $vs$ 621.6M). This performance also validates our two-stage design, as Stage 2 significantly enhances the reconstruction quality from Stage 1 to reach this state-of-the-art level. This demonstrates that AliTok can maintain great reconstruction performance while simultaneously creating a generation-friendly token sequence.

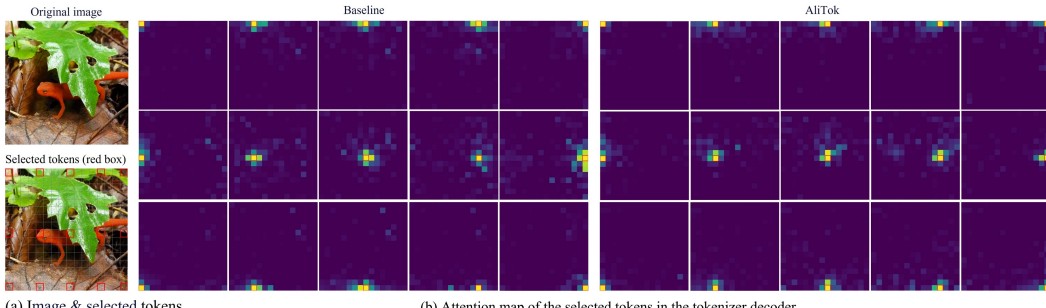

(a) Image & selected tokens     (b) Attention map of the selected tokens in the tokenizer decoder

Figure 11: **More attention maps for selected tokens** in the bidirectional decoder. In contrast to the symmetric attention distribution of the baseline, AliTok's attention exhibits a strong causal bias (concentrating on the top-left), even within a fully bidirectional decoder.

**Visualization of more attention maps.** To confirm the generality of the attention pattern revealed in the main text, we provide additional attention maps in Fig. 11 for individual tokens at various spatial locations, particularly from the edges and the center of the image. For the baseline model, its bidirectional decoder widely references the global context during reconstruction, with attention freely dispersed across both causal history and future regions. In contrast, the attention in AliTok's bidirectional decoder is heavily skewed towards drawing information from historical tokens, largely ignoring the available context from future regions. This indicates that the necessary contextual information for each token has been effectively organized within its causal history, thereby facilitating the subsequent unidirectional generation process.

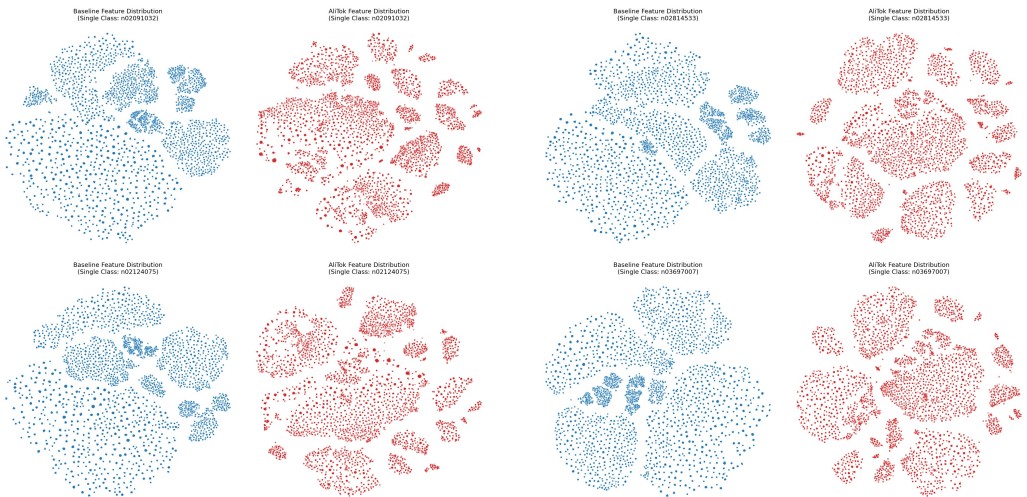

Figure 12: **t-SNE visualization of codebook vectors.** The plot compares the feature distributions of codebook vectors used for a single ImageNet class, generated by the baseline ViT-VQ tokenizer (blue) and AliTok (red).

**Visualization of codebook.** We performed a t-SNE analysis to visualize the codebooks learned by a standard ViT-VQ (conventional approach in our paper) and AliTok. We encoded 200 images from a single ImageNet class with each tokenizer, and collected the utilized codebook vectors for 2D projection. The resulting visualization in Fig. 12 shows a clear topological divergence. The baseline's feature space (blue) is diffuse and continuous, characterized by ambiguous cluster boundaries and significant feature entanglement. This suggests that the unconstrained bidirectional encoder creates non-causal dependencies across tokens to achieve high reconstruction fidelity. AliTok (red),

by contrast, learns a well-structured discrete manifold of compact, clearly separated semantic clusters. This demonstrates that AliTok's causal decoder constraint forces the encoder to learn more deterministic and disentangled representations. This structured and predictable feature distribution presents a more learnable target for the autoregressive model, which partly explains the significant improvement in generation accuracy observed with AliTok.

Table 11: KV-Caching Ablation (256×256, w/ cfg, batch=64).

| Setting | Sampling Time (batch=64) | Speedup |
|---|---|---|
| AliTok-XL (w/o KV-Cache) | 107.76s | 1.0× |
| AliTok-XL (w/ KV-Cache) | **10.07s** | **10.7×** |

Table 12: Sampling cost at different resolutions.

| Tokenizer | 256×256 (batch=64, w/ cfg) | | | 512×512 (batch=64, w/ cfg) | | |
|---|---|---|---|---|---|---|
| | Tokens | Sampling time | Memory | Tokens | Sampling time | Memory |
| Baseline-XL | 256 | 9.20s | 14.81G | 1024 | 95.17s | 46.98G |
| AliTok-XL | 272 | 10.07s | 15.48G | 1056 | 100.67s | 48.33G |
| Δ | (+6.3%) | (+9.5%) | (+4.5%) | (+3.1%) | (+5.8%) | (+2.9%) |

**Detailed Analysis of Sampling Efficiency and Scalability.** we provide a comprehensive analysis of the sampling efficiency of AliTok, explaining the primary driver behind its significant speed advantage and detailing the scalability of our approach with respect to the prefix tokens. First, AliTok's 10× speedup over masked models and diffusion-based models primarily stems from its full compatibility with Key-Value (KV) caching. In sequential generation, KV-caching stores past states to avoid re-computation at each step. Our ablation study in Table 11 confirms its dominant role: disabling KV-caching skyrockets inference time from 10.07s to 107.76s.

Regarding scalability, we further quantified the overhead of additional prefix tokens at higher resolutions. As shown in Table 12, the relative impact of these tokens decreases significantly as resolution increases. For instance, scaling from 256×256 to 512×512 reduces the extra sampling time overhead from 9.5% to 5.8%. This favorable trend occurs because prefix tokens grow linearly (O(W)) while total tokens grow quadratically (O(W×H)), making the overhead asymptotically negligible.

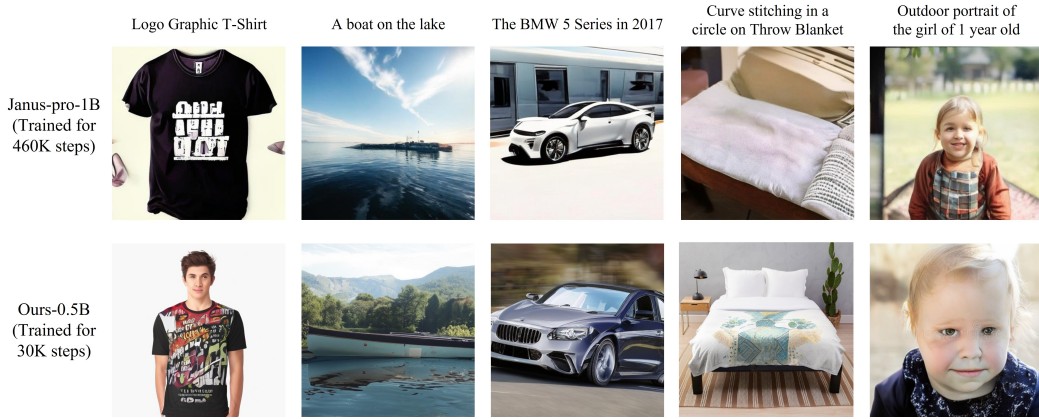

Figure 13: **Comparison of text-to-image generation between Janus-pro-1B and ours-0.5B.** Note that our model was trained for merely 30K steps.

**Generalization to Multimodal Text-to-Image Generation.** To investigate AliTok's generalization capability, we conducted a preliminary text-to-image experiment. Following the unified paradigm of Janus-pro (Chen et al., 2025), we built a 0.5B parameter model where a single autoregressive (AR) architecture handles both text understanding and visual generation. This model is based on LLaVA-OV-0.5B (Li et al., 2024a) and incorporates AliTok as its visual tokenizer. The model was trained

on the CC12M dataset (Changpinyo et al., 2021) at 512×512 resolution for only 30,000 steps on 64 A100 GPUs.

Fig. 13 shows a qualitative comparison against the 1B-parameter Janus-pro model (trained for 460K steps). Despite the significant disparity in model scale and training duration, our model demonstrates the ability to generate coherent images corresponding to the text prompts. This result serves as an initial validation of AliTok's effectiveness and efficiency in a challenging multimodal autoregressive setting, highlighting its potential for broader applications.

### A.5    USE OF LARGE LANGUAGE MODELS (LLMS)

In accordance with ICLR 2026 policy, we report our use of a large language model (LLM) during the preparation of this manuscript. The LLM's role was strictly confined to language polishing, such as correcting grammar and refining word choices. The authors are solely responsible for all scientific contributions, including the ideation, methodology, experimental design, and final conclusions. The authors have reviewed and take full responsibility for the final manuscript.

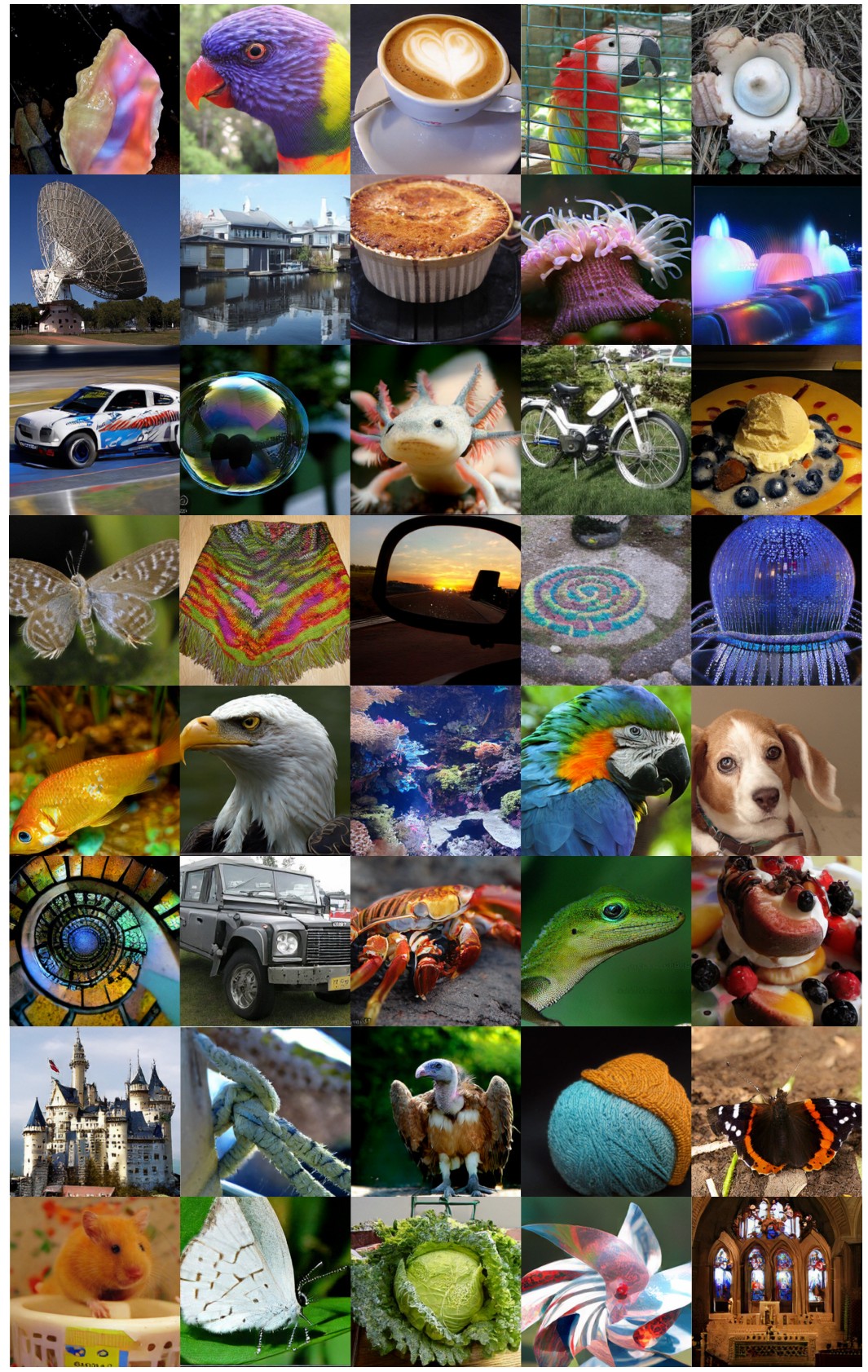

Figure 14: **More generation results** on the ImageNet 256×256 benchmark using our AliTok-XL.

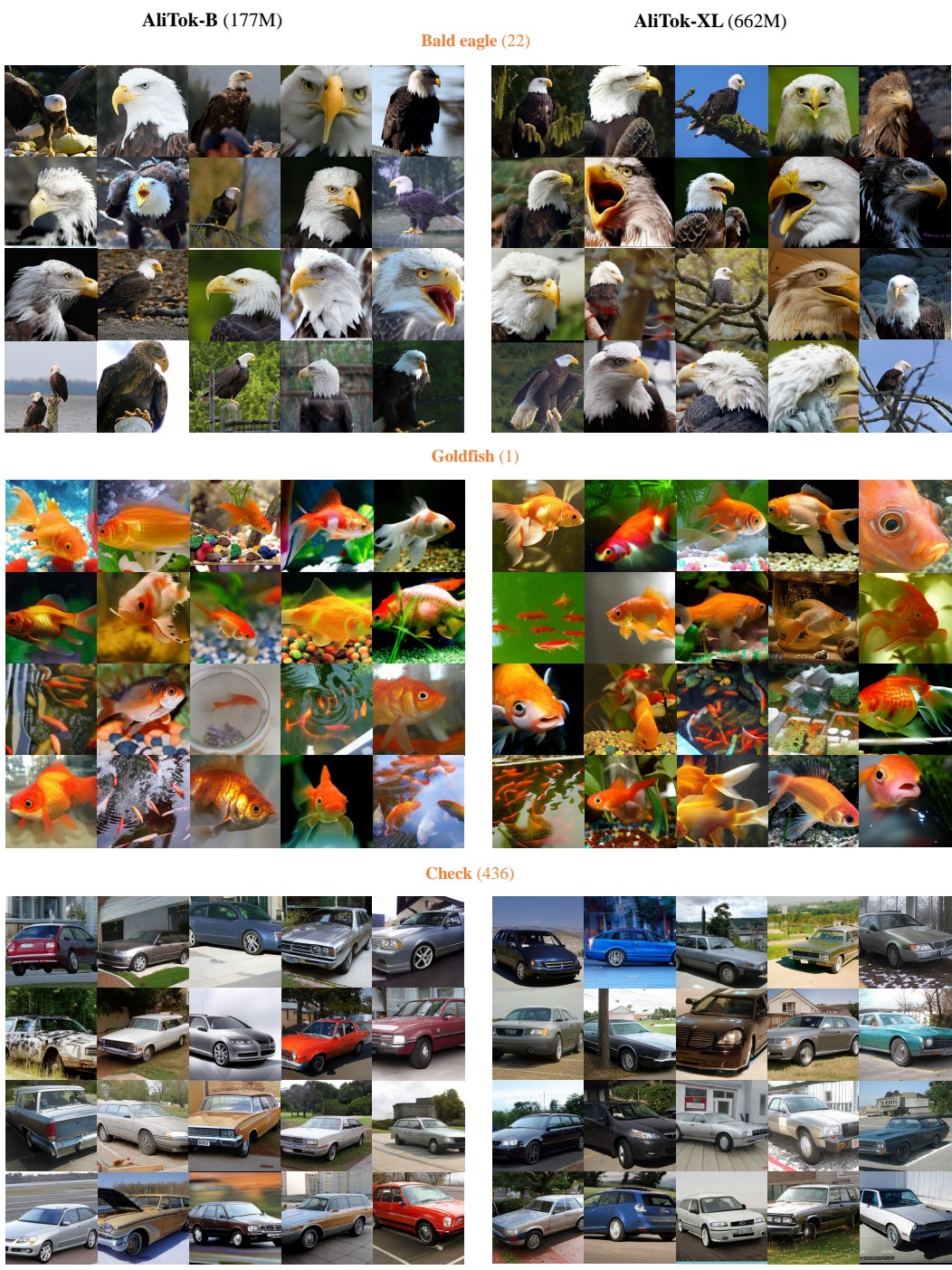

Figure 15: **More generation results.** The left images are generated by AliTok-B (177M), and the right images are generated by AliTok-XL (662M).

**AliTok-B** (177M)          **AliTok-XL** (662M)

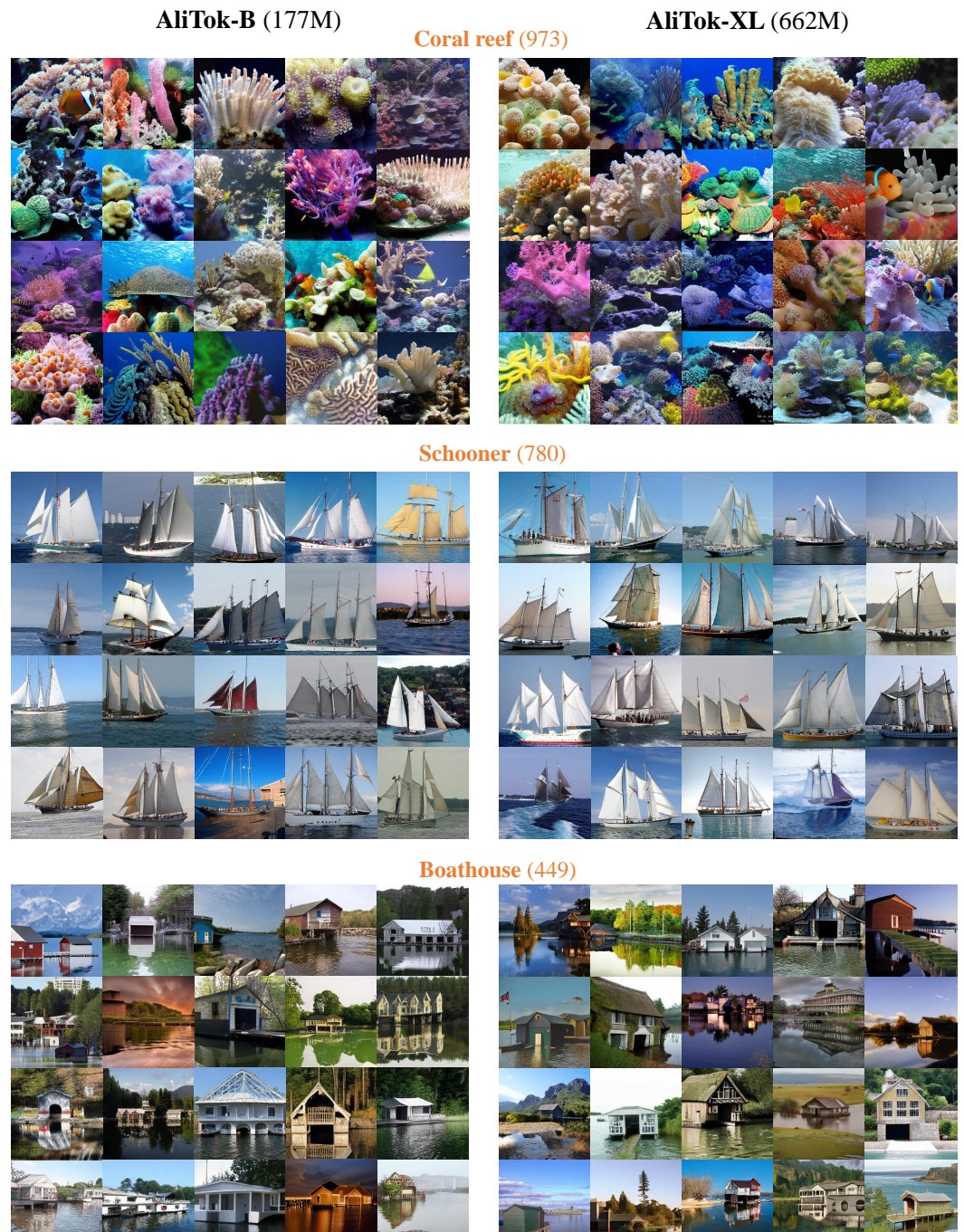

Figure 16: **More generation results.** The left images are generated by AliTok-B (177M), and the right images are generated by AliTok-XL (662M).

**AliTok-B** (177M)          **AliTok-XL** (662M)

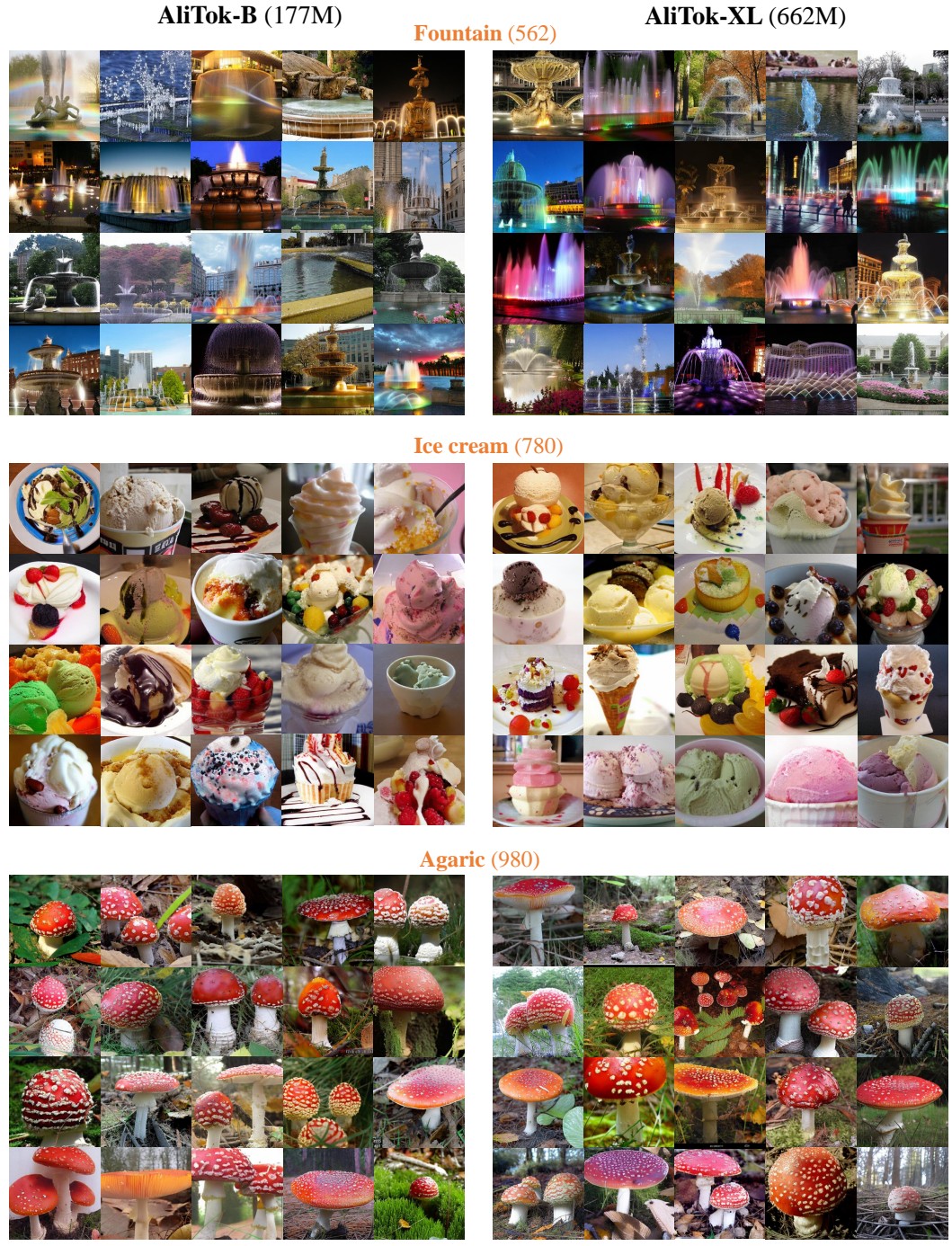

Figure 17: **More generation results.** The left images are generated by AliTok-B (177M), and the right images are generated by AliTok-XL (662M).

**AliTok-B** (177M)       **AliTok-XL** (662M)

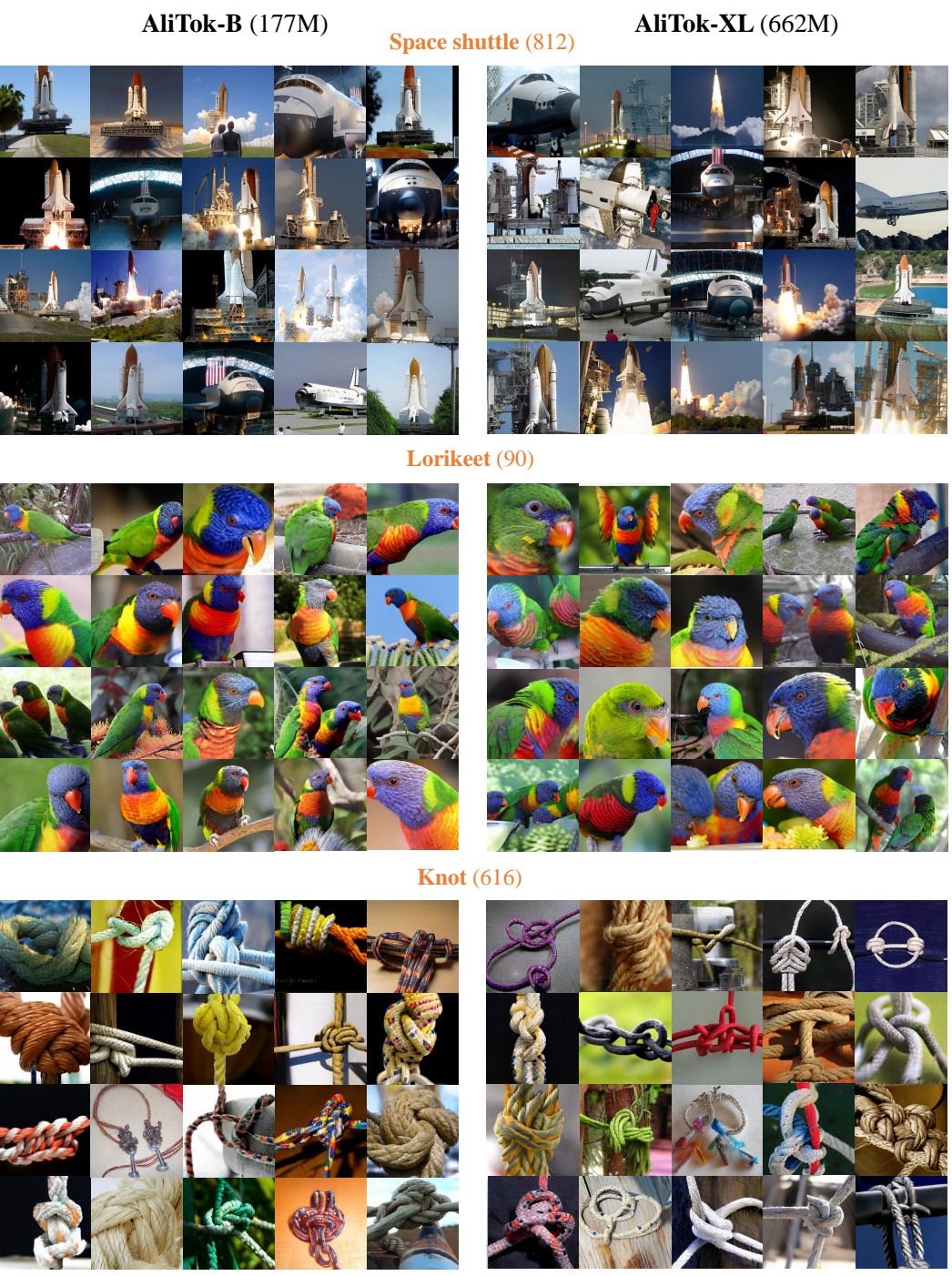

Figure 18: **More generation results.** The left images are generated by AliTok-B (177M), and the right images are generated by AliTok-XL (662M).

