# OpenReview forum: "Towards Sequence Modeling Alignment between Tokenizer and Autoregressive Model"
_ICLR.cc/2026/Conference — ICLR 2026 Poster_

### Official Review · Reviewer_yYv4 · 2025-10-19

**Soundness:** 3
**Presentation:** 3
**Contribution:** 3
**Rating:** 6
**Confidence:** 4

**Summary:**

This paper tackles an fundamental issue in visual autoregressive modeling: the mismatch between the bidirectional structure of visual tokenizers and the unidirectional nature of autoregressive models. The authors propose AliTok, a new tokenizer that introduces a causal decoder during training, enforcing the encoder to produce tokens with forward-dependency patterns more aligned with autoregressive modeling. A two-stage training strategy and prefix tokens are used to balance generation-friendliness and reconstruction fidelity.

**Strengths:**

1. The motivation of the paper is clearly articulated and the proposed approach is effective.
2. The paper is well-organized and clearly written. The framework is described in detail and the authors provide sufficient information about it.
3. Instead of modifying the AR model, the authors redesign the tokenizer training objective to produce causally-aligned tokens. This “tokenizer alignment” perspective is novel and could generalize to other modalities.

**Weaknesses:**

1. AliTok forces tokens to be easier to predict, but that very constraint reduces representational richness.
2. It seems that there is no guarantee that the learned codebook remains causally coherent after AR fine-tuning.

**Questions:**

1. Can you visualize or quantify how the learned codebook differs from standard VQGAN/ViT-VQ tokens?
2. How does AliTok affect token entropy compared to standard bidirectional tokenizers? A measurable drop would suggest over-regularization, the model becomes easier to train but less expressive.

---

> ### Author Response · Authors · 2025-11-22
>
> **[W1] Reduction in representational richness**:
>
> Thank you for this insightful question. Our two-stage training process is explicitly designed to decouple and balance the trade-off between predictability and representational richness.
>
> In the first stage, We use a causal decoder as a regularizer to force the bidirectional encoder to learn a generation-friendly latent structure. While this constraint leads to a slight degradation in reconstruction performance (rFID deteriorates from 0.98 to 1.07), this is **partially attributable to the limitations of the causal decoder itself**. To address this, we freeze the generation-friendly encoder and codebook, and retrain a powerful bidirectional decoder. **This second stage allows our final rFID improve from 0.91 to 0.86**.
>
> Ultimately, our final model employs a bidirectional encoder and decoder, realizing significantly enhanced predictability with only a marginal loss in representational richness compared to the baseline.
>
>
> **[W2] Causally Coherent After AR Fine-tuning**:
>
> We appreciate you raising this important concern. We wish to clarify a key aspect of our training protocol: **after the first stage of training, the weights of both the encoder and the codebook are explicitly frozen**. This design preserves the causally coherent structure learned in the first stage. Consequently, all subsequent training processes, including the training of the AR model and the bidirectional decoder, do not alter the encoder or the codebook. This approach fundamentally precludes the concern you raised.
>
> **[Q1] Visualization of Codebook**:
>
> Thank you for this constructive suggestion. We have visualized the difference between the AliTok and standard ViT-VQ (baseline) codebooks through a t-SNE analysis. The resulting comparison plot and detailed discussion are provided in Appendix (Fig. 12).
>
> The visualization reveals a distinct contrast in the topological structure of the feature spaces. The standard ViT-VQ features exhibit large, diffuse regions with blurred boundaries between semantic clusters, indicating that the unconstrained bidirectional encoder tends to learn highly entangled representations. In sharp contrast, AliTok's features form a highly structured and discrete manifold, characterized by compact, well-separated clusters with sharp semantic boundaries. This demonstrates that AliTok's causal constraint effectively forces the encoder to learn representations with sharply disentangled, fine-grained semantic specificity, leading to clearer separation. We believe this disentanglement partly explains the observed improvement in generation accuracy.
>
> Furthermore, while t-SNE intuitively illustrates the static geometric differences, it is less effective at capturing dynamic inter-token dependencies. Therefore, we also provide an Attention Map visualization analysis in the main paper (Fig. 8). This complementary analysis directly confirms, from the perspective of sequence dependency, that AliTok's tokens exhibit a significantly stronger causal bias.
>
>
> **[Q2] Impact on Token Entropy and Model Expressiveness**:
>
> Thank you for this insightful question. We wish to clarify that AliTok aims to reduce the conditional entropy that hinders autoregressive prediction, not to decrease model expressiveness.
>
> Conventional bidirectional tokenizers, under an unconstrained reconstruction objective, tend to create representational dependencies on future tokens, which **inevitably inflates the conditional entropy $p(z_i|z_{<i})$**.
>
> On the other hand, directly employing a strict causal encoder constitutes a form of over-regularization. While this hard constraint ensures an intrinsic causal dependency, **its limited receptive field impairs semantic extraction**, thereby reducing representational richness.
>
> In contrast, AliTok strikes an optimal balance. We retain a bidirectional encoder to ensure all tokens possess rich semantics derived from a global receptive field, thus preserving expressive power. Simultaneously, we introduce a causal decoder constraint to compel the encoder to optimize information flow, eliminating representational dependencies on non-causal tokens within the compressed sequence, thereby enhancing sequence predictability. This design, bidirectional encoding with causal constraints, achieves a critical alignment of the dependency structure while maintaining high information density, rather than sacrificing feature richness for predictability. This conclusion is also confirmed by our competitive reconstruction metrics.

---

> > ### Comment · Reviewer_yYv4 · 2025-11-25
> >
> > Thanks for your reply, I will keep my score.

---

### Official Review · Reviewer_5Mwz · 2025-10-25

**Soundness:** 3
**Presentation:** 3
**Contribution:** 3
**Rating:** 6
**Confidence:** 4

**Summary:**

This paper tackles the misalignment between conventional bidirectional image tokenizers and unidirectional autoregressive (AR) models in image generation by introducing AliTok, an aligned tokenizer. AliTok uses a bidirectional encoder constrained by a causal decoder to produce semantically rich, forward-dependent token sequences, plus prefix tokens and a two-stage training process to boost reconstruction and generation performance. On ImageNet-256, AR models with AliTok excel: a 177M-parameter model achieves 1.44 gFID and 319.5 IS, while the 662M-parameter AliTok-XL hits 1.28 gFID  with 10× faster sampling.

**Strengths:**

1. The paper is clearly written and well-structured, making it easy to read and understand.
2. The work provides a compelling analysis of the fundamental conflict between conventional bidirectional tokenizers and unidirectional autoregressive (AR) models, and addresses it through a novel design that combines a bidirectional encoder with a causally constrained decoder to facilitate AR image generation.
3. The proposed framework is thoroughly validated through extensive experiments, ablation studies, and well-motivated design choices, including the use of a causal decoder, prefix tokens with an auxiliary loss, and a two-stage tokenizer training strategy.
4. The proposed AliTok approach achieves state-of-the-art performance on the ImageNet-256 benchmark, surpassing leading diffusion-based methods while offering significantly faster sampling speed.

**Weaknesses:**

1. The experimental evaluation is limited to ImageNet-256; results on higher-resolution datasets such as ImageNet-512 are missing, which leaves uncertainty about the scalability and robustness of the proposed method to larger image resolutions.
2. The paper does not provide a detailed analysis or explanation for the reported 10× sampling speedup. My understanding is that the method introduces additional buffer tokens and prefix tokens, which could potentially incur higher computational cost compared to the baseline. A clear breakdown of the factors contributing to the acceleration would strengthen the claims.
3. The approach may face challenges when scaling to higher resolutions, as the number of prefix tokens is likely to grow with image size (e.g., for ImageNet-512), which might impact efficiency and memory usage.

**Questions:**

See weakness.

---

> ### Author Response · Authors · 2025-11-22
>
> **[W1] Results at ImageNet-512**:
>
> Thank you for your valuable suggestion. Following your recommendation, we have extended AliTok to the 512×512 ImageNet class-conditional generation task. Remarkably, by fine-tuning our pretrained AliTok-L model at 512×512 resolution for only 60 epochs, **we achieved a gFID of 1.39, significantly surpassing the previous state-of-the-art method, MAR (gFID=1.73)**. Detailed experimental procedures and analyses have been added to Section 4.4 of the revised paper (specifically Table 5 and Figure 9). Figure 9 visually demonstrates the excellent generation quality achieved by our model at this higher resolution. Furthermore, we commit to open-sourcing the code and weights for these experiments as well.
>
> | Method    | #params | gFID↓ |  IS↑ |
> |-----------|---------|-------|-------|
> | REPA      | 675M    | 2.08  | 274.6 |
> | MAR-L     | 481M    | 1.73  | 279.9 |
> | AliTok-L  | 318M    |**1.39** | **295.3**|
>
>
> **[W2] Analysis for Sampling Speedup**:
>
> Thank you for this important question. The 10× sampling speedup is primarily due to AliTok adopting a standard autoregressive paradigm that is natively compatible with KV-caching, while simultaneously enabling performance comparable to SOTA diffusion models with similar parameter counts. To quantify this, we conducted an ablation study where disabling KV-caching caused the sampling time to increase significantly from 10.07s to 107.76s, confirming it as the fundamental driver of the acceleration. Regarding the overhead from extra tokens you noted, our experiments show the impact is marginal. The additional prefix tokens incur a 9.5% increase in sampling latency, while the buffer tokens (operating solely within the tokenizer decoding) impact the speed by less than 1%. These extra costs are overall limited compared to the substantial efficiency gains realized. We have integrated this detailed computational analysis into the revised Appendix (Table 11) for clarity.
>
> | Setting    | Sampling Time (batch=64) | Relative Speedup |
> |-----------|---------|-------|
> | AliTok-XL (w/o KV-Cache)   | 107.76s    | 1.0x |
> | AliTok-XL (w/ KV-Cache)   | 10.07s | 10.7x|
> | | |
> | w/o Prefix Tokens | **9.20s** | **11.7x** |
> | w/o Buffer Tokens | 10.05s | 10.7x |
>
>
> **[W3] Prefix Token Overhead at Scale**:
>
> We appreciate your concern regarding scalability to higher resolutions. We have conducted thorough experiments to directly quantify this impact.
>
> | Batch=64 |  |  256x256 | || | 512x512  ||
> |-|-|-|-|-|-|-|-|
> | Tokenizer | Tokens | Sampling time | Memory || Tokens | Sampling time  | Memory |
> | Baseline-XL  | 256 | 9.20s | 14.81G | | 1024 | 95.17s |  46.98G |
> | AliTok-XL  | 272 | 10.07s | 15.48G || 1056 | 100.67s | 48.33G |
> | Extra Cost  | +6.3% | +9.5% | +4.5% | | +3.1% | +5.8% | +2.9%  |
>
> As shown in the table, when scaling from 256×256 to 512×512, the relative overhead decreases significantly for both sampling time (9.5%→5.8%) and GPU memory (4.5%→2.9%). This favorable trend exists because the number of prefix tokens only linearly with image width (O(W)), whereas the total tokens scales quadratically with the area (O(WxH)). Thus, the relative proportion of prefix tokens diminishes as resolution increases, rendering the additional cost asymptotically negligible and confirming the scalability of our approach.

---

> > ### Comment · Reviewer_5Mwz · 2025-11-27
> >
> > Thanks for the comprehensive response, I will keep my score.

---

### Official Review · Reviewer_u2Nd · 2025-10-31

**Soundness:** 2
**Presentation:** 3
**Contribution:** 2
**Rating:** 4
**Confidence:** 4

**Summary:**

The paper proposes AliTok, an image tokenizer aimed at reducing the misalignment between conventional bidirectional visual tokenization and the unidirectional nature of autoregressive (AR) models.
AliTok trains a bidirectional encoder under a causal decoder constraint, encouraging the latent tokens to exhibit forward-dependency so that an AR decoder can better model them. The authors further introduce prefix tokens for first-row reconstruction and a two-stage training scheme to restore image fidelity.
On ImageNet-256, AliTok enables a standard AR transformer to outperform diffusion models (gFID 1.28 vs 1.35) and sample about 10× faster.

**Strengths:**

- Clear motivation and writing. The authors communicate the misalignment problem and causal-decoder idea clearly.

- Comprehensive ablations. The study isolates each design component’s effect on AR accuracy and gFID.

- Interpretability. Attention-map visualization nikens learn a causal bias.

- Empirical competence. Experiments are reproducible, metrics are standard, and odel sizes.

**Weaknesses:**

- **Limited conceptual novelty:**
  The core idea—adding a causal constraint to the tokenizer—is elegant but incremental, extending existing causal-structure ideas (e.g., masked or random-order AR) rather than offering a fundamentally new paradigm.

- **Order-specific and brittle design:**
  The method hard-codes raster order and needs ad-hoc fixes (prefix tokens, auxiliary loss) for top-row reconstruction. It is unclear if the alignment holds under different scan orders, resolutions, or tasks.

- **Two-stage compromise undermines the claim:**
  The encoder is trained causally but later paired with a bidirectional decoder, partially reversing the intended unidirectional alignment.

- **Narrow experiments:**
  Results are limited to ImageNet-256 class-conditional generation. No tests on higher resolutions, text-conditional, or multimodal setups—so generality is unproven.

- **Marginal quantitative gains and unclear fairness:**
  The gFID improvement over diffusion (1.35 → 1.28) is small and possibly within evaluation noise; speedups rely on different kernel and caching settings without matched baselines.

- **Tokenizer bottleneck:**
  The discrete tokenizer itself caps achievable FID (~1.15), limiting scalability unless codebook size or representation capacity increases.

**Questions:**

While raster-scan decoding is the traditional form of autoregressive image generation, it is widely recognized as suboptimal for visual data due to its rigid ordering and limited global context.
Recent works such as **MaskGIT**, **MAR** have successfully relaxed this constraint using masked or random-order prediction, enabling bidirectional context while maintaining autoregressive structure.

Given this progress, why does your method **intentionally return to a strict raster-scan decoding scheme** instead of adopting or extending these more flexible paradigms?
Doesn’t enforcing raster causality reintroduce the very inefficiencies and spatial myopia that modern AR methods have been trying to overcome?
How can you justify this design choice as a step forward rather than a regression in autoregressive modeling?

---

> ### Author Response · Authors · 2025-11-22
>
> **[W1] Limited Conceptual Novelty**:
>
> We thank the reviewer for acknowledging the elegance of our idea. We would like to clarify that AliTok represents a **fundamental conceptual shift** rather than an incremental extension. Previous works, such as Masked/Random-order AR, share a common philosophy of **adapting the model to the data**, addressing the inherent bidirectional dependencies in images through complicating the model architecture. In contrast, AliTok explores the opposite path of **adapting the data to the model**. We reshape the dependency structure of the visual sequence at the tokenizer level, enabling it to natively fit the purest form of the autoregressive paradigm.
>
> Our experiments demonstrate that once the data and the model are conceptually aligned, a standard autoregressive model is sufficient to surpass all previous complex AR methods, and even rival SOTA diffusion models. AliTok’s core contribution is **identifying and resolving this fundamental mismatch**, thereby restoring the competitiveness of standard autoregressive modeling.
>
>
>
> **[W2] Order-specific and Brittle Design**:
>
> Thank you for raising this important point. We would like to offer the following clarifications:
>
> Choice of Raster Order: We focus on raster-scan due to its **native compatibility with the left-to-right generation logic of LLMs**. Unlike other ordering schemes, this unique characteristic provides a seamless, unified modeling approach for future multimodal generation, which is a primary motivation for our work.
>
> Generalizability Across Orders: While implementation components like prefix tokens are tailored for the raster order, AliTok’s **core alignment principle applies to any predefined scanning sequence** (e.g., spiral in order), as the unidirectional nature of the AR model remains constant regardless of the specific order.
>
> Robustness Across Resolutions and Tasks: **We have extended our method to ImageNet-512** during the rebuttal (Figure 9 and Table 5), achieving a state-of-the-art gFID of 1.39, which verifies its robustness in higher resolutions. To further verify task-generality at high resolution, **we have conducted a preliminary multimodal experiment at 512x512 resolution** by integrating AliTok into a 0.5B LLaVA-OV model. As detailed in Appendix A.4 (Fig. 13), it shows promising results after only 30K training steps. This provides strong initial evidence that the standard AR architecture powered by AliTok can generalize to high-resolution multimodal tasks.
>
>
> **[W3] Two-stage Compromise Undermines the Claim**:
>
> Thank you for this question, which allows us to clarify the core idea behind our two-stage design. We argue this is not a compromise, but an effective decoupling of objectives. The purpose of the first stage is to shape the encoder and codebook, forcing them to learn a representation with forward dependencies via a causal decoder as a powerful structural regularizer. Critically, we **completely freeze the well-trained encoder and codebook in the second stage**. This action ensures that the generation-friendly properties from the first stage cannot be altered. On this foundation, we independently train a more powerful bidirectional decoder to achieve maximum reconstruction fidelity. Therefore, our two-stage training strategy does not reverse the unidirectional alignment of the encoded sequence.
>
>
> **[W4] Narrow Experiments**:
>
> We understand your concerns regarding generalizability. Following your suggestion, we **have extended AliTok-L to ImageNet-512** class-conditional generation task (Figure 9 and Table 5) and achieved a **gFID of 1.39**, surpassing the previous SOTA method MAR-L (gFID=1.73). This strongly demonstrates that our tokenizer design can robustly scale to higher resolutions.
>
> | Method    | #params | gFID↓ |  IS↑ |
> |-----------|---------|-------|-------|
> | REPA      | 675M    | 2.08  | 274.6 |
> | MAR-L     | 481M    | 1.73  | 279.9 |
> | AliTok-L  | 318M    |**1.39** | **295.3**|
>
> Regarding task-generality and high-resolution performance, we appreciate your suggestion and have made an effort to address both concerns. The results of this new experiment are now included in Appendix A.4. Following the unified paradigm of Janus-pro, **we integrated AliTok as the visual tokenizer into a 0.5B LLaVA-OV model**. Due to the time and computational constraints of the rebuttal period, this model was trained at 512x512 resolution for only 30K steps on the CC12M dataset. Despite these significant limitations, the model already demonstrates a promising ability to generate plausible images from text prompts (Fig. 13), serving as a strong proof-of-concept for AliTok's generalization to high-resolution multimodal autoregressive settings. We believe this preliminary result validates the potential of our approach, and we are committed to conducting more extensive experiments for the final version of the paper.

---

> ### Author Response · Authors · 2025-11-22
>
> **[W5] Marginal Quantitative Gains and Unclear Fairness**:
>
> We thank the reviewer for ensuring the rigor of our comparisons. To rule out evaluation noise, we re-evaluated AliTok-XL across five random seeds. The resulting mean gFID is 1.2835 ± 0.0048 (individual scores: 1.2917, 1.2804, 1.2835, 1.2821, 1.2799). The tight error margin confirms that our improvement is reliable. Crucially, **within the autoregressive paradigm, this represents a massive leap** over the previous SOTA, RAR-XL (1.48 gFID), marking a fundamental rather than incremental advance.
>
> Regarding the fairness of the speed comparison, the KV-caching we employ is a standard optimization practice for all autoregressive models, including the approaches we compare (e.g., RAR, VAR). **This ensures a fair comparison within the AR paradigm**. As for the comparison with Diffusion, we ensured a fair benchmark by using officially provided, highly-optimized implementations for all methods. Therefore, our speed advantage stems from a paradigm-level difference, rather than from unfair implementation techniques.
>
>
> **[W6] Tokenizer Bottleneck**:
>
> We share the reviewer's perspective. To validate the scalability of our approach in addressing this bottleneck, we **conducted an additional experiment by increasing the codebook size from 4096 to 16384**. The comparison results have been added to Table 6 of the paper and are presented below:
>
> | Codebook Size  | rFID (val set) | rFID (train set) | gFID↓     | IS↑       | Loss↓     |   Acc↑    |
> |--------------- |----------------|------------------|-----------|-----------|-----------|-----------|
> | 4096           | 0.86           | 1.15             | **1.28**  | **306.3** | **4.67**  | **12.2%** |
> | 16384          | **0.81**       | **1.09**         | 1.30      | 304.9     | 6.42      |   4.3%    |
>
> Using a larger codebook effectively lifts the reconstruction upper bound, confirming the scalability of the AliTok tokenizer. However, the expanded vocabulary space leads to a sharp drop in training accuracy, resulting in slightly inferior generation performance for our XL-sized AR model with the 16384 codebook compared to the 4096 codebook. We anticipate that with extended training epochs and increased model parameters, larger codebooks will achieve better generation performance, and we leave this exploration to future work.
>
>
> **[Q] Significance of Raster-scan AR Paradigm**:
>
> Thank you for this profoundly insightful question. We would like to clarify our design philosophy.
>
> We agree that flexible-ordering paradigms like MaskGIT and MAR are significant advances, successfully mitigating the spatial myopia of traditional AR. However, they introduce modality-specific complexity into modeling and sampling. Our work explores the opposite, fundamental question: Rather than altering the proven, scalable standard AR model (GPT-style) to fit visual sequence, can we transform the sequence itself (via better tokenization) to seamlessly integrate into this unified paradigm? **This is precisely why we intentionally return to strict raster-scan order: to maintain absolute consistency with the standard LLM approach for a unified paradigm.**
>
> In response to your question, "How can you justify this as a step forward rather than a regression?", we believe that **solving complex problems with simpler principles is a form of advancement**. Our work demonstrates three key points:
> 1. It is not necessary to continually complicate model architectures for visual tasks. An aligned tokenizer makes the simplest scan-order powerful enough to model visual structures.
> 2. The simplest AR paradigm's potential is far from exhausted. Past underperformance was due to inadequate sequence representation, not architectural flaws.
> 3. This "path of regression" is not only feasible but also superior. The combined power of an aligned tokenizer and a simple AR model surpasses more complex architectures, including SOTA diffusion models.
>
> Therefore, we firmly believe that successfully re-integrating a specialized problem (visual generation) back into a general framework (standard AR) and demonstrating its superior performance **is not a "regression"**. On the contrary, it is a powerful demonstration of the core principle of building more general, simple, and powerful unified models.

---

### Official Review · Reviewer_ubw6 · 2025-11-01

**Soundness:** 3
**Presentation:** 3
**Contribution:** 3
**Rating:** 6
**Confidence:** 5

**Summary:**

This paper proposes AliTok, a discret image tokenizer designed to match the unidirectional nature of autoregressive models. AliTok is trained in a two-stage manner, the first stage uses causal decoder to regularize the encoder, and the second stage uses bidirectional decoder to improve reconstruction. AliTok also employs prefix tokens to fix the "first row" issue in autoregressive models. AliTok achieves comparable or better performance on ImageNet dataset with a faster sampling speed.

**Strengths:**

1. The intuition of tailoring the tokenizer to suit the AR generation is well-established. Image as 2-d grid does not really have a naturally direction. Several existing method uses random order AR. AliTok solves this with 1d tokenizer and a causal decoder.
2. The fix for "first row" using prefix tokens is smart. As stated, image does not really have direction, this makes the first row hard to predict. Using prefix row to encode global semantic information provides guidance for this with minimal cost (256 + 17 prefix).
3. The experiments results demonstrate the superior generation performance with extensive ablations and visualization.

**Weaknesses:**

1. No system-level comparison of reconstruction performance. Despite the final goal for tokenizer is to enable better generation, a detailed comparison and analysis of reconstruction performance cannot be neglected. This is a huge missing in a paper focusing on tokenizer.
2. Several experimental details are missing, especially for reconstruction. For example, what is the batch size / epochs used in two-stage training? What are the size of the tokenizer compared to other model? Without those details, it is hard for apples-to-apples comparison.
3. Sampling speed is not really a vital contribution of AliTok. It naturally fits in autoregressive models which originates from LlamaGen.

**Questions:**

1. A system-level comparison of reconstruction performance is a must for this paper. Especially with codebook size / codebook dim / training epochs / number of parameters.
2. In Tab. 2, the paper uses original codebases for all methods. Does this mean other methods do not use KV-Cache?
3. In Tab. 2, the performance of GigaTok is attributed to its codebook size. How does AliTok perform with different vocabulary size?

---

> ### Author Response · Authors · 2025-11-22
>
> **[W1, Q3] Systematic Reconstruction Comparison**:
>
> We appreciate the reviewer's constructive feedback. We agree that a systematic reconstruction comparison is crucial for a tokenizer-focused paper. Following your excellent suggestion, we have included a comprehensive evaluation of reconstruction metrics in the revised Appendix (Table 10). More importantly, to ensure a direct, apples-to-apples comparison with GigaTok, **we have trained a new AliTok variant with a matched codebook size of 16384**. The results are summarized below. Our AliTok (Stage 2) achieves highly comparable performance to the state-of-the-art GigaTok-B-L across all key metrics (PSNR, SSIM, LPIPS, and rFID). Thanks to your suggestion, our now more comprehensive evaluation strongly validates that our two-stage design achieves top-tier reconstruction fidelity.
>
> | Tokenizer        | Tokens | Codebook | PSNR↑   | SSIM↑   | LPIPS↓  | rFID↓  | #Params |
> |------------------|--------|----------|---------|---------|---------|--------|---------|
> | TiTok-L-32       | 32     | 4096     | 16.15   | 0.5098  | 0.3232  | 2.21   | 641.1M  |
> | TiTok-B-64       | 64     | 4096     | 17.39   | 0.5491  | 0.2539  | 1.70   | 204.8M  |
> | AliTok (Stage 1) | 272    | 4096     | 20.66   | 0.6571  | 0.1659  | 0.91   | 389.8M  |
> | AliTok (Stage 2) | 272    | 4096     | **20.97**   | **0.6610**  | **0.1573**  | **0.86**   | 390.0M  |
> | | | | | | | | |
> | GigaTok-B-L      | 256    | 16384    | 21.24   | **0.6873**  | **0.1284**  | **0.81**   | 621.6M  |
> | AliTok (Stage 1) | 272    | 16384    | 21.08   | 0.6805  | 0.1367  | 0.84   | 390.2M  |
> | AliTok (Stage 2) | 272    | 16384    | **21.30**   | 0.6851  | 0.1303  | **0.81**   | 390.4M  |
>
>
> **[W2, Q1] Tokenizer Training Specifications**:
>
> We thank the reviewer for pointing out the missing details. To provide the necessary context for a systematic and fair comparison, we have now added a comprehensive table with full training configurations in the revised Appendix (Table 8).
>
> As shown in the table, our comparisons are conducted under fair conditions. While we adopted the same batch size and learning rate, AliTok utilizes a moderate parameter count (390M), substantially smaller than GigaTok-B-L (622M) and TiTok-S (641M). Furthermore, our total training duration of 180 epochs is slightly less than the 200 epochs used by others. We trust this added context provides the necessary clarity regarding the fairness of our system-level comparison.
>
> | Specification         | TiTok-B | TiTok-S | GigaTok-B-L | AliTok (Stage 1) | AliTok (Stage 2) |
> |-----------------------|---------|---------|-------------|------------------|------------------|
> | Codebook Size         | 4096    | 4096    | 16384       | 4096             | 4096             |
> | Codebook Dimension    | 12      | 12      | 8           | 32               | 32               |
> | Training epochs       | 200     | 200     | 200         | 120              | 60               |
> | #Parameters           | 204.8M  | 641.1M  | 621.6M      | 389.8M           | 390.0M           |
> | | | | | | |
> | Total Batch Size      | 256     | 256     | 256         | 256              | 256              |
> | Base Learning Rate    | 1e-4    | 1e-4    | 1e-4        | 1e-4             | 1e-4             |
> | Minimum Learning Rate | 1e-5    | 1e-5    | 1e-5        | 1e-5             | 1e-5             |
>
>
> **[W3] Sampling Speed Contribution**:
>
> We fully agree with the reviewer's insightful point. Fast sampling is indeed an inherent advantage of the AR paradigm, not a direct contribution of our work.
>
> Our key contribution is not inventing this advantage, but making it practically viable at the SOTA performance level. For years, this speed advantage was of little practical significance, as standard AR models could not match the generation quality of top diffusion models, even at much larger parameter scales. AliTok resolves the core sequence-model misalignment and closes this quality gap. It is this breakthrough in generation quality that **makes the inherent speed advantage of AR models truly meaningful and practical in a SOTA-level comparison**.
>
>
> **[Q2] KV-Cache Usage in Speed Comparison**:
>
> Thank you for this important question, as it concerns a crucial detail for a convincing evaluation. To uphold this principle of fairness, we utilized the original codebases specifically to ensure that each method runs in its official, optimized environment provided by the authors. We can confirm the following regarding KV-Cache usage:
> 1. For autoregressive models like **VAR and RAR, KV-Cache is supported and was explicitly enabled in our evaluation**.
> 2. Non-causal methods such as MAR and LightningDiT inherently do not support KV-Cache.
>
> Therefore, our evaluation protocol ensures a valid and fair comparison, and Table 2 accurately reflects the inference speed differences resulting from the intrinsic architectures of each method.

---

### Author Response · Authors · 2025-11-22

We sincerely thank all reviewers for their constructive feedback. We have incorporated your feedback throughout the main text and appendix. Beyond the reviewer-specific changes, we also highlighted two major experimental additions:

1. **ImageNet-512 SOTA Results**: We extended AliTok-L to 512×512 resolution (Figure 9 and Table 5), achieving a state-of-the-art gFID of 1.39, significantly surpassing the previous best method (MAR-L, 1.73).
2. **Large Codebook Experiment**: We added a new AliTok variant with a 16384 codebook size, demonstrating reconstruction fidelity on par with SOTA tokenizers like GigaTok, and provided an initial exploration of its impact on generative training  (Table 6 and Table 10).

---

### Meta-Review · Area_Chair_Bs2k · 2026-01-14

**Summary:**

This paper argues that standard image tokenizers bring about bidirectional dependencies in the token sequence, and it is misaligned with the unidirectional factorization used by decoder-only autoregressive (AR) models. To address this, the authors propose AliTok, which trains a bidirectional encoder under a causally constrained decoder to encourage forward-dependent tokens, and uses prefix tokens plus a two-stage tokenizer training procedure to improve both predictability and reconstruction. Empirically, AliTok enables strong class-conditional ImageNet generation.

**Reviewer Concerns:**

Addressed:
- Missing systematic reconstruction comparison / missing tokenizer training details (ubw6): The authors added a detailed reconstruction table (PSNR/SSIM/LPIPS/rFID) and trained an AliTok variant with a matched 16384 codebook for head-to-head comparison, plus provided training specs (epochs, batch size, params, etc.)
- Speedup explanation + prefix/buffer overhead and scaling to 512 (5Mwz): The authors attributed the speedup primarily to KV-cache, provided an ablation (w/ vs w/o KV-cache), and quantified overhead from prefix/buffer tokens. They also reported ImageNet-512 and measured relative overhead scaling.
- Representational richness, codebook causal coherence (yYv4): The authors clarify the two-stage design and state that encoder+codebook are frozen after stage 1; they also discuss the predictability–expressiveness trade-off.
- Narrow experiments / scaling to higher resolution (u2Nd, 5Mwz): Added ImageNet-512 class-conditional results.

Outstanding / partially addressed:
- Conceptual novelty (u2Nd): The authors provide a philosophical framing (“adapt data to model” vs “adapt model to data”), but this may still read as an incremental variant of “causalizing” tokenization.
- “Evaluation noise / marginal gain” skepticism (u2Nd): The authors report a 5-seed re-evaluation for gFID with a small error bar. "marginal gains" concern is still valid.
- Generality beyond ImageNet class-conditional (u2Nd): The rebuttal includes a preliminary 512×512 multimodal experiment (0.5B LLaVA-OV, 30K steps), but it remains a proof-of-concept; stronger evidence (more training, more datasets/tasks, and/or text-to-image benchmarks) would better support broad claims.
- Codebook scaling trade-off (u2Nd): While the authors explore a larger codebook and show improved reconstruction upper bound, the generation metrics slightly degrade and training accuracy drops, suggesting scaling behavior still needs deeper study.

**Reviewer Scores:**

- Reviewer ubw6 (score: 6 → likely 6, possibly 7)
- Reviewer u2Nd (score: 4 → likely 4–5)
- Reviewer 5Mwz (score: 6 → 6)
- Reviewer yYv4 (score: 6 → 6)

---

### Decision · Program_Chairs · 2026-01-26

Accept (Poster)